# DATA-CENTRIC PREDICTION EXPLANATION VIA KERNELIZED STEIN DISCREPANCY

**Mahtab Sarvmaili, Hassan Sajjad, Ga Wu**
Department of Computer Science
Dalhousie University
{mahtab.sarvmaili,hsajjad,ga.wu}@dal.ca

## ABSTRACT

Existing example-based prediction explanation methods often bridge test and training data points through the model's parameters or latent representations. While these methods offer clues to the causes of model predictions, they often exhibit innate shortcomings, such as incurring significant computational overhead or producing coarse-grained explanations. This paper presents a Highly-precise and Data-centric Explanation (HD-Explain) prediction explanation method that exploits properties of Kernelized Stein Discrepancy (KSD). Specifically, the KSD uniquely defines a parameterized kernel function for a trained model that encodes model-dependent data correlation. By leveraging the kernel function, one can identify training samples that provide the best predictive support to a test point efficiently. We conducted thorough analyses and experiments across multiple classification domains, where we show that HD-Explain outperforms existing methods from various aspects, including 1) preciseness (fine-grained explanation), 2) consistency, and 3) computation efficiency, leading to a surprisingly simple, effective, and robust prediction explanation solution. Source code is available at https://github.com/MahtabSarvmaili/HDExplain.

## 1 INTRODUCTION

As one of the decisive factors affecting the performance of a Machine Learning (ML) model, training data points are of great value in promoting the model's transparency and trustworthiness, including explaining prediction results, tracing sources of errors, or summarizing the characteristics of the model (Cai et al., 2019; Anik & Bunt, 2021; Nam et al., 2022). The challenges of example-based prediction explanation mainly come from retrieving relevant data points from a vast pool of training samples or justifying the rationale of such explanations (Lim et al., 2019; Zhou et al., 2021).

Modern example-based prediction explanation methods commonly approach the above challenges by constructing an influence chain between training and test data points (Li et al., 2020; Nam et al., 2022; Tsai et al., 2023). The influence chain could be either data points' co-influence on model parameters or their similarity in terms of latent representations. In particular, Influence Function (Koh & Liang, 2017), one of the representative, model-aware explanation methods, looks for the shift of the model parameters (due to up-weighting each training sample) as the sample's influence score. Since computing the inverse Hessian matrix is challenging, the approach adapts Conjugate Gradients Stochastic Estimation and the Perlmutter trick to reduce its computation cost. Representer Point Selection (RPS) (Yeh et al., 2018), as another example, reproduces the representer theorem by refining the trained neural network model with $L2$ regularization, such that the influence score of each training sample can be represented as the gradient of the predictive layer. While computationally efficient, RPS is criticized for producing coarse-grained explanations that are more class-level rather than instance-level explanations (Sui et al., 2021) (In this paper we use the instance level explanation and example-based explanation interchangeably.). Multiple later variants (Pruthi et al., 2020; Sui et al., 2021) attempted to mitigate the drawbacks above, but their improvements were often limited by the cause of their shared theoretical scalability bounds.

This paper presents Highly-precise and Data-centric Explanation (HD-Explain), a post-hoc, model-aware, example-based explanation solution for neural classifiers. Instead of relying on data co-

Table 1: Summary of existing Post-hoc Example-based Prediction Explanation Methods that work with deep neural networks. Practicality of the whole model explanation is measured by the feasibility of explaining the prediction of ResNet-18 trained on CIFAR-10 with a single A100 GPU machine. CIFAR-10 is a small benchmark data with 50000 training samples.

| Method | Explanation of | Need optimization as sub-routine | Whole model explanation | | Inference computation complexity bounded by | Memory/cache (**of each training sample**) bounded by |
|---|---|---|---|---|---|---|
| | | | Theoretical | Practical | | |
| Influence Function | Original Model | Yes (Iterative HVP approximation) | Yes | No | $1. H_\theta^{-1} \nabla_\theta L(\mathbf{x}_t, \theta)$ approximation $2. < \nabla_\theta L(\mathbf{x}, \theta), H_\theta^{-1} \nabla_\theta L(\mathbf{x}_t, \theta) >$ | Size of model parameters |
| RPS | Fine-tuned Model | Yes (L2 regularized last layer retrain) | No | No | $1.$last layer representation $\mathbf{f}_t$ $2. < \alpha_i \mathbf{f}_i, \mathbf{f}_t >$ | Size of model parameters of the last layer |
| TracIn* | Original Model | No | Yes | No | $1. \nabla_\theta L(\mathbf{x}_t, \theta)$ approximation $2. < \nabla_\theta L(\mathbf{x}, \theta), \nabla_\theta L(\mathbf{x}_t, \theta) >$ | Size of model parameters |
| HD-Explain | Original Model | No | Yes | Yes | $1. \nabla_{\mathbf{x}_t} f(\mathbf{x}_t, \theta)_{y_t}$ 2. Closed-form $k_\theta(\mathbf{x}, \mathbf{x}_t)$ defined by KSD | Size of data dimension |

* TracIn typically requires to access the training process. Here, TracIn* refers to a special case that only use the last training checkpoint.

influence on model parameters or feature representation similarity, HD-Explain retains the influence chain between training and test data points by exploiting the underrated properties of Kernelized Stein Discrepancy (KSD) (Liu et al., 2016) between the trained predictive model and its training dataset. Specifically, we note that the Stein operator augmented kernel uniquely defines a pairwise data correlation (in the context of a trained model) whose expectation on the training dataset results in the minimum KSD (as a discrete approximation) compared to that of the dataset sampled from different distributions. By exploiting this property, we can 1) reveal a subset of training data points that provides the best predictive support to the test point and 2) identify the potential distribution mismatch among training data points. Jointly leveraging these advantages, HD-Explain can produce explanations faithful to the original trained model.

The contributions of our work are summarized as follows: 1) We propose a novel example-based explanation method. 2) We propose several quantitative evaluation metrics to measure the correctness and effectiveness of generated explanations. 3) We perform a thorough evaluation comparing several existing explanation methods across a wide set of classification tasks.

Our findings conclude that HD-Explain offers fine-grained, instance-level explanations with remarkable computational efficiency, compared to well-known example-based prediction explanation methods. Its algorithmic simplicity, effective performance and scalability enables its deployment to real world scenarios where transparency and trustworthiness is essential.

## 2 PRELIMINARY AND RELATED WORK

### 2.1 POST-HOC CLASSIFIER EXPLANATION BY EXAMPLES

Post-hoc Classifier Explanation by Examples (a.k.a prototypes, representers) refers to a category of classifier explanation approaches that pick a subset of training data points as prediction/explanations without accessing the model training process. Its research history spans from the model-intrinsic approach (see survey Molnar (2020)) to the recent impact-based approach (Li et al., 2020).

Model-inherent approaches (Molnar, 2020) refer to machine learning models that are considered interpretable such as $k$-nearest neighbor (Peterson, 2009) or decision tree; For a given test data point, similar data points on the raw feature space can be efficiently selected as explanations through the inherent decision making mechanism of the self-explanatory machine learning models. In fact, attracted by their inherent explanatory power, multiple well-known works attempted to compile complex black-box models into self-explanatory models for enabling prediction explanation (Frosst & Hinton, 2017), while computationally inefficient.

To unlock the general explanatory power applicable to black-box models, multiple later studies suggested to fall back to statistics-based solutions, looking for prototype samples that represent data points that either are common in the dataset or play critical roles in data distribution. MMD-critic (Kim et al., 2016) and Normative and Comparative explanations (Cai et al., 2019) are the well-known examples in this category. Unfortunately, those approaches are often with strong assumption of *good*

prototypes (which often overlook the characteristics of trained models) (Li et al., 2020), making their prediction explanations general to the training dataset rather than a trained model instance.

Recently, influence-based methods have emerged as the prevailing technique in model explanation (Li et al., 2020; Nam et al., 2022; Bae et al., 2022; Park et al., 2023a). Influence function (Koh & Liang, 2017), as one of the earliest influence-based solutions, bridges the outcome of a prediction task to training data points by, first, evaluating training data's influence on the model parameters and, then, estimating how model parameter changes affect prediction. Similarly, Representative Point Selection (RPS) (Yeh et al., 2018) builds such an influence chain by fitting the representation theorem, where the weighted product between the representations of test and training samples comes into play. Concerning the computational overhead of previous work, the later solution TracIn (Pruthi et al., 2020) proposed a simple approximation of the influence function via a first-order Taylor expansion (essentially Neural Tangent Kernel (Jacot et al., 2018)), successfully discarding the inverse Hessian matrix from the influence chain formulation. BoostIn (Brophy et al., 2023) further extends TracIn and is dedicated to interpreting the predictions of gradient-boosted decision trees. RPS-LJE (Sui et al., 2021), on the other hand, alleviated the inconsistent explanation problem of RPS through Local Jacobian Expansion. In the latest publication (Tsai et al., 2023), all of the above methods described in this paragraph are identified as special cases of *Generalized Representers* but with different chosen kernels.

One limitation of the current influence-based methods is that they attribute the influence of each training data point to the parameters of the trained model as an essential intermediate step. Indeed, as the nature of stochastic gradient descent (the dominating training strategy of neural networks), isolating such contribution is barely possible without 1) relying on approximations or 2) accessing the training process. Unfortunately, either solution would result in performance degradation or heavy computational overhead (Schioppa et al., 2022). Hence, this work delves into the exploration of an alternative influence connection between training and test data points without exploiting the perturbation of model parameters.

## 2.2 KERNELIZED STEIN DISCREPANCY

The idea of Kernelized Stein Discrepancy (KSD) (Liu et al., 2016) can be traced back to a theorem called Stein Identity (Kattumannil, 2009) that states, if a smooth distribution $p(x)$ and a function $\phi(x)$ satisfy $\lim_{||x|| \to \infty} p(x)\phi(x) = 0$,

$$\mathbb{E}_{x \sim p}[\phi(x)\nabla_x \log p(x) + \nabla_x \phi(x)] = 0, \quad \forall \phi.$$

The identity can characterize distribution $p(x)$ such that it is often served to assess the goodness-of-fit (Kubokawa, 2024) of the model. The above expression could be further abstracted to use function operator $\mathcal{A}_p$ (a.k.a Stein operator) such that

$$\mathcal{A}_p \phi(x) = \phi(x)\nabla_x \log p(x) + \nabla_x \phi(x),$$

where the operator encodes distribution $p(x)$ in the form of derivative to input (a.k.a score function).

Stein's identity offers a mechanism to measure the gap between two distributions by assuming the variable $x$ is sampled from a different distribution $q \neq p$ such that

$$\sqrt{\mathbb{S}(q, p)} = \max_{\phi \in \mathcal{F}} \mathbb{E}_{x \sim q}[\mathcal{A}_p \phi(x)],$$

where the expression takes the most discriminant $\phi$ that maximizes the violation of Stein's identity to quantify the distribution discrepancy. This discrepancy is, accordingly, referred as Stein Discrepancy.

The challenge of computing Stein Discrepancy comes from the selection of function set $\mathcal{F}$, which motivates the later innovation of KSD that takes $\mathcal{F}$ to be the unit ball of a reproducing kernel Hilbert space (RKHS). By leveraging the reproducing property of RKHS, the KSD could be eventually transformed into

$$\mathbb{S}(q, p) = \mathbb{E}_{x, x' \sim q}[\kappa_p(x, x')]$$

where $\kappa_p(x, x') = \mathcal{A}_p^x \mathcal{A}_p^{x'} k(x, x')$ that can work with arbitrary kernel function $k(x, x')$. See Appendix C for expanded derivations.

In the literature, KSD has been adopted for tackling three types of application tasks – 1) parameter inference (Barp et al., 2019), 2) Goodness-of-fit tests (Chwialkowski et al., 2016; Liu et al., 2016;

Figure 1: Varying of Kernelized Stein Discrepancy given the shift of training data distribution on Two Moon dataset.

Yang et al., 2018), and 3) particle filtering (sampling) (Gorham et al., 2020; Korba et al., 2021). However, to the best of our knowledge, its innate property that uniquely defines model-dependent data correlation has never been exploited, which, we note, is valuable to interpret model behaviour from various aspects, including instance-level prediction explanation and global prototypical explanations.

## 3 HIGHLY-PRECISE AND DATA-CENTRIC EXPLANATION

HD-Explain is an example-based prediction explanation method based on Kernelized Stein Discrepancy. Consider a trained classifier $f_\theta$ as the outcome of a training process with Maximum Likelihood Estimation (MLE)

$$\operatorname*{argmax}_{\theta} \mathbb{E}_{(\mathbf{x},y) \sim P_D}[\log P_\theta(y|\mathbf{x})].$$

Theoretically, maximizing observation likelihood is equivalent to minimizing a KL divergence between data distribution $P_D$ and the parameterized distribution $P_\theta$ such that

$$\mathbb{D}_{\mathrm{KL}}(P_D, P_\theta) = \mathbb{E}_{(\mathbf{x},y) \sim P_D}\left[\log \frac{P_D(\mathbf{x},y)}{P_\theta(\mathbf{x},y)}\right]$$

$$= -\underbrace{\mathbb{E}_{(\mathbf{x},y) \sim P_D}[\log P_\theta(y|\mathbf{x})]}_{\text{Likelihood}} + \underbrace{\mathbb{E}_{(\mathbf{x},y) \sim P_D}[\log P_D(y|\mathbf{x})]}_{\text{constant}} + \underbrace{\mathbb{E}_{(\mathbf{x},y) \sim P_D}[\log P_D(\mathbf{x}/P_\theta(\mathbf{x}))]}_{\text{constant as } \theta \text{ does not model inputs}},$$

which, in turn, is proven to align with minimizing KSD in the form of gradient descent (Liu & Wang, 2016)

$$\nabla_\epsilon \mathbb{D}_{\mathrm{KL}}(P_D, P_\theta)|_\epsilon = -\mathbb{S}(P_D, P_\theta),$$

where $\epsilon$ is the step size of gradient decent.

The chain of reasoning above shows that a well-trained classifier $f_\theta$ through gradient-descent should lead to minimum discrepancy between the training dataset distribution and the model encoded distribution $\mathbb{S}(P_D, P_\theta)$ [1]. We empirically verify the connection through simple examples as shown in Figure 1, where the changes in training data distribution would result in larger KSD compared to that of the original training data distribution. Intuitively, the connection shows that there is a tie between a model and its training data points, encoded in the form of a Stein kernel function $k_\theta(\cdot, \cdot)$ defined on each pair of data points. As the kernel function is conditioned on model $f_\theta$, we note it is an encoding of data correlation under the context of a trained model, which paves the foundation of the example-based prediction explanation.

### 3.1 KSD BETWEEN MODEL AND TRAINING DATA

Recall that KSD, $\mathbb{S}(P_D, P_\theta)$, defines the correlation between pairs of training samples through model $\theta$ dependent kernel function with closed-form decomposition

$$\kappa_\theta((\mathbf{x}_a, y_a), (\mathbf{x}_b, y_b)) = \mathcal{A}_\theta^a \mathcal{A}_\theta^b k(a, b)$$
$$= \nabla_a \nabla_b k(a, b) + k(a, b) \nabla_a \log P_\theta(a) \nabla_b \log P_\theta(b) \qquad (1)$$
$$+ \nabla_a k(a, b) \nabla_b \log P_\theta(b) + \nabla_b k(a, b) \nabla_a \log P_\theta(a),$$

where we denote data point $(\mathbf{x}_a, y_a)$ with $a$ for clean notation. The only model-dependent factor in the above decomposition is a derivative $\nabla_{\mathbf{x},y} \log P_\theta(\mathbf{x}, y)$ (for both data $a$ or $b$).

---

[1] Since $P_D$ is discrete distribution while $P_\theta$ is continuous, the Discrepancy between the two distributions will not recap Stein Identity ($= 0$) with a limited number of training data points.

However, as the KSD only models the discrepancy between joint distributions rather than conditional distributions, it is challenging to estimate discrepancy between predictive models $P_\theta(y|\mathbf{x})$ and its training set $\mathbf{z} = (\mathbf{x}, y) \sim P_D$ without including $P_\theta(\mathbf{x})$ into consideration, even though the marginal distribution $P_\theta(\mathbf{x})$ is not estimated by the predictive model at all.

Inspired by the previous study on Goodness of Fit (Jitkrittum et al., 2020), to unlock KSD support on predictive models, we propose to set $P_\theta(\mathbf{x}) \equiv P_D(\mathbf{x})$ such that the identical marginal distribution would not contribute to the discrepancy between the joint distributions $P_D(\mathbf{x}, y)$ and $P_\theta(\mathbf{x}, y)$. In addition, while original distribution $P(\mathbf{x})$ that generates data could be arbitrary complex distribution (that is out of modelling scope of the predictive model), we may simply set data point distribution $P_D$ (not original distribution $P$) as an Uniform distribution over data points in the dataset under the particular context, given a data point is sampled from a generated dataset uniformly. Although the relaxation appears hasty, we believe that the relaxation is valid in the prediction explanation context (and probably only valid in this particular context) in terms of measuring correlation between a pair of data points, where all train/test data points' input are valid observation (e.g. images) rather than random continuous valued sample in the same space form an unknown distribution.

With the above relaxations, the score function $\nabla_{\mathbf{x},y} \log P_\theta(\mathbf{x}, y)$ in the Stein operator $\mathcal{A}_\theta$ could be derived as a concatenation of the gradient of model $f_\theta(\mathbf{x})_y$ to its input $\mathbf{x}$ and its probabilistic prediction $f_\theta(\mathbf{x})$ in logarithm form, since

$$
\begin{aligned}
\nabla_{\mathbf{x},\mathbf{y}} \log P_\theta(\mathbf{x}, \mathbf{y}) &= \nabla_{\mathbf{x},\mathbf{y}} [\log P_\theta(\mathbf{y}|\mathbf{x}) + \log P_D(\mathbf{x})] \\
&= \nabla_{\mathbf{x},\mathbf{y}} \log P_\theta(\mathbf{y}|\mathbf{x}) + [\nabla_{\mathbf{x}} \log P_D(\mathbf{x}) || \nabla_{\mathbf{y}} P_D(\mathbf{x})] \\
&= [\nabla_{\mathbf{x}} \mathbf{y}^\top \log f_\theta(\mathbf{x}) || \nabla_{\mathbf{y}} \mathbf{y}^\top \log f_\theta(\mathbf{x})] + [\mathbf{0} || \mathbf{0}] \\
&= [\nabla_{\mathbf{x}} \log f_\theta(\mathbf{x})_y || \log f_\theta(\mathbf{x})],
\end{aligned}
\tag{2}
$$

where $[\cdot || \cdot]$ denotes concatenation operation. We use one-hot vector representation $\mathbf{y}$ to represent data label $y$ here. Since $P_D(\mathbf{x})$ follows uniform distribution, its gradient to the inputs is a 0 vector.

In the above derivation, we treat discrete label $y$ as-is without specialized discrete distribution treatments (see (Yang et al., 2018)) to avoid significant computation overhead. In fact, data space in practice is unlikely dense even if a group of features are continuous (e.g. images). Treating the label as a sparse continuous feature can be viewed as an approximation.

Combining Equation 1 and 2, we can estimate the correlation of any pairs of training data points conditioned on the trained machine learning model. Computationally, since a score function $\nabla_{\mathbf{x},y} \log P_\theta(\mathbf{x}, y)$ depends on a single data point, its outputs of the training set could be pre-computed and cached to accelerate the kernel computation. In particular, the output dimension of the score function is simply $m + k$ for data with $m$ dimensional features and $k$ class labels. Compared to the existing solutions, whose training data cache (or influence) are bounded by the dimension of model parameters (such as Influence function, TracIn, RPS, RPS-JLE), the explanation method built on KSD would come with a significant advantage in terms of scalability (see comparison in Table 1). This statement is generally true for neural network based classifiers, whenever the size of model parameters is far larger than the data dimension.

## 3.2 Prediction Explanation

The computation of kernel function in Equation 1 requires access to features and labels of a data point. While the ground-truth label information is available for the training set, it is inaccessible for a test data point. We consider the predicted class $\hat{y}_t$ of the test data point $\mathbf{x}_t$ as a label to construct a complete data point $(\mathbf{x}_t, \hat{y}_t)$ and apply the KSD kernel function. For a test data point $\mathbf{x}_t$, we search for top-k training data points that maximize the KSD defined kernel.

Figure 2 demonstrates HD-Explain on a 2d synthetic dataset. The distribution of $\kappa_\theta(\mathbf{d}, \cdot)$ in the right most plot shows that only a small number of training data points have a strong influence on a particular prediction.

## 4 Evaluation and Analysis

In this section, we conduct several qualitative and quantitative experiments to demonstrate various properties of HD-Explain and compare it with the existing example-based solutions.

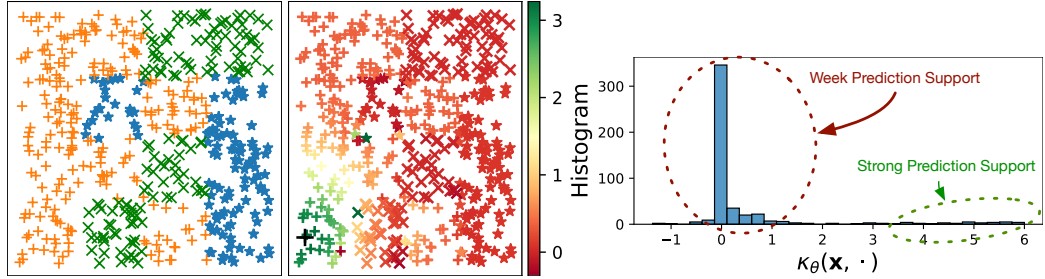

Figure 2: Demonstration of HD-Explain on 2D Rectangular synthetic dataset. Left shows the training dataset with three classes. Middle figure shows the explanation support of training data points to a given test point (as black cross), where green shows a higher KSD kernel value. Right shows the distribution of KSD kernel values (over the training set) to the test point, where only a small number of training data points provide strong support to this prediction.

**Datasets:** We consider multiple disease classification tasks where diagnosis explanation is highly desired. We also introduced synthetic and benchmark classification datasets to deliver the main idea without the need for medical background knowledge. Concretely, we use CIFAR-10 ($32 \times 32 \times 3$), Brain Tumor (Magnetic Resonance Imaging, $128 \times 128 \times 3$), Ovarian Cancer (Histopathology Images, $128 \times 128 \times 3$) datasets, and SVHN ($32 \times 32 \times 3$). More details are listed in the Appendix F.

**Baselines:** The baseline explainers used in our experiments include Influence Function, Representer Point Selection, and TracIn. While other variants of these baseline explainers exist (Barshan et al., 2020; Sui et al., 2021; Chen et al., 2021), we note they don't offer fundamental performance improvements over the classic ones. In addition, as Influence Function and TracIn face scalability issues, we limit the influence of parameters to the last layer of the model so that they can work with models that contain a large number of parameters. Our experiments use ResNet-18 as the backbone model architecture (with around 11 million trainable parameters) for all image datasets (see Appendix H for detail on our hardware setup).

Finally, we also introduce an HD-Explain variant (HD-Explain*) to match the last layer setting of other baseline models, even though HD-Explain can scale up to the whole model without computation pressure. The HD-Explain* is a simple change of HD-Explain in terms of using data representations (the output of the last non-predictive layer of the neural classifier) rather than the raw features. Specifically, we assume a neural network model $f_\theta$ could be decomposed into two components $f_{\theta_2} \cdot f_{\theta_1}$, where $f_{\theta_1}$ is a representation encoder and $f_{\theta_2}$ is a linear model for prediction. With this decomposition, we define the KSD kernel function for HD-Explain* as

$$
\begin{aligned}
\kappa_\theta((f_{\theta_1}(\mathbf{x}_a), y_a), (f_{\theta_1}(\mathbf{x}_b), y_b)) &= \mathcal{A}_{\theta_2}^a \mathcal{A}_{\theta_2}^b k(a, b) \\
&= \nabla_a \nabla_b k(a, b) + k(a, b) \nabla_a \log P_{\theta_2}(a) \nabla_b \log P_{\theta_2}(b) \\
&\quad + \nabla_a k(a, b) \nabla_b \log P_{\theta_2}(b) + \nabla_b k(a, b) \nabla_a \log P_{\theta_2}(a),
\end{aligned}
$$

where we define $a = (f_{\theta_1}(\mathbf{x}_a), y_a)$ and $b = (f_{\theta_1}(\mathbf{x}_b), y_b)$ for short. This setting reduces the prediction explanation to the last layer of the neural network in a similar fashion to RPS.

**Metrics:** In existing example-based explanation works, the experimental results are often demonstrated qualitatively, as visualized explanation instances, without quantitative evaluation. This results in subjective evaluation. In this paper, we propose several quantitative evaluation metrics to measure the effectiveness of each method.

- **Hit Rate:** Hit rate measures how likely an explanation sample hits the desired example cases where the desired examples are guaranteed to be undisputed. Specifically, we modify a training data point with minor augmentations (adding noise or flipping horizontally) and use it as a test data point, such that the best explanation for the generated test data point should be the original data point in the training set.

- **Coverage:** Given n test data points, the metric measures the number of unique explanation samples an explanation method produces when configuring to return top-k training samples.

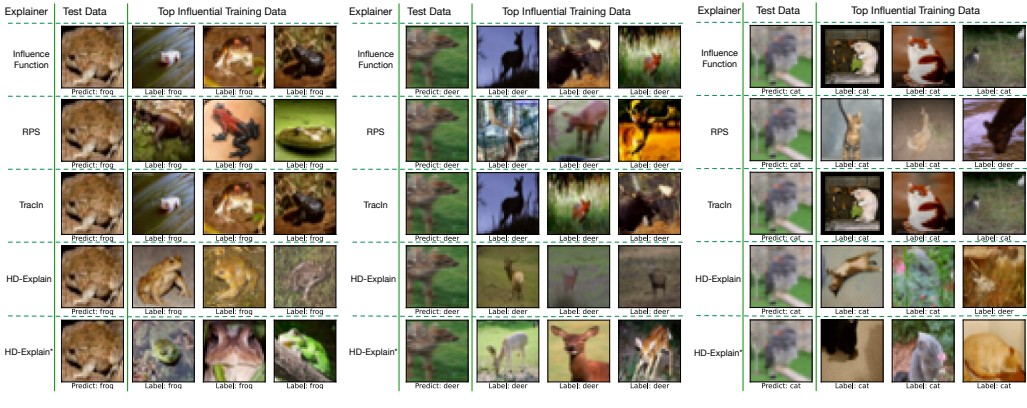

(a) High-Conf Correct Predict     (b) Low-Conf Correct Predict     (c) Low-Conf Incorrect Predict

Figure 3: Qualitative evaluation of various example-based explanation methods using CIFAR10. We show three scenarios where the target model makes a) a highly-confident prediction that matches ground truth label, b) a low-confident prediction that matches ground truth label, c) low-confident prediction that does not match ground truth label (which is a bird). For each sub plot, we show top-3 influential training data points picked by the explanation methods for the test example.

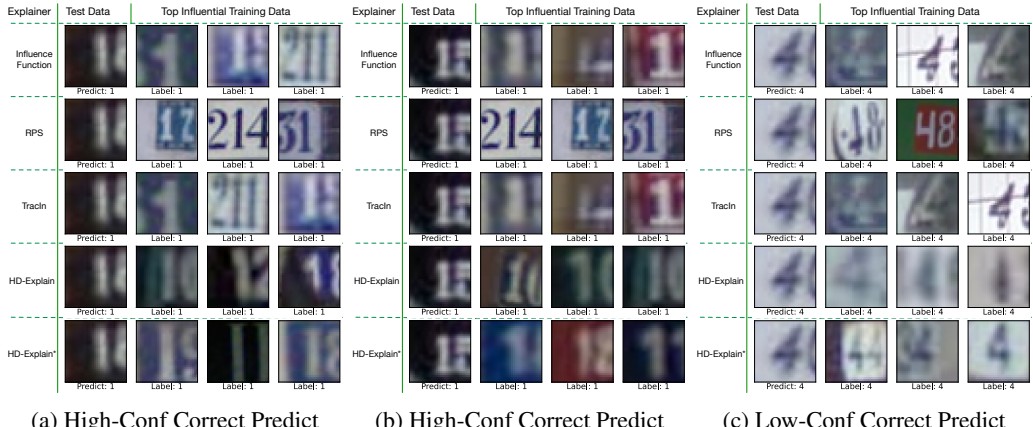

(a) High-Conf Correct Predict     (b) High-Conf Correct Predict     (c) Low-Conf Correct Predict

Figure 4: Qualitative evaluation of various example-based explanation methods using SVHN. We show two scenarios where the target model makes a-b) a highly-confident prediction that matches ground truth label, c) a low-confident prediction that matches ground truth label. For each sub plot, we show top-3 influential training data points picked by the explanation methods for the test example. We include two samples of high-confidence correct predictions to show the overlap of explanations.

Formally,

$$\text{Coverage} = \frac{|\cup_{i=1}^{n} e_i|}{n \times k}$$

where $e_i$ is the set of top-k explanations for test data point $i$. Coverage is motivated to measure the diversity of explanations across a test set where a high value reflects higher granularity (per test point) of the explanation.

• **Run Time:** It measures the run time of an explanation method in wall clock time.

## 4.1 QUALITATIVE EVALUATION

Figure 3 shows three test cases of the CIFAR10 classification task that cover different classification outcomes, including high-confidence correct prediction, low-confident correct prediction, and low-confident incorrect prediction. For both correct prediction cases, we are confident that HD-Explain provides a better explanation than others in terms of visually matching test data points e.g. brown

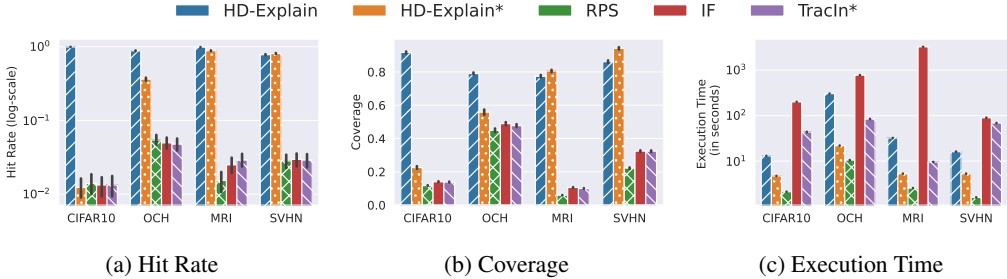

(a) Hit Rate        (b) Coverage        (c) Execution Time

Figure 5: Quantitative explanation comparison among candidate example-based explanation methods. Data augmentation strategy used is Noise Injection. Error bar shows 95% confidence interval.

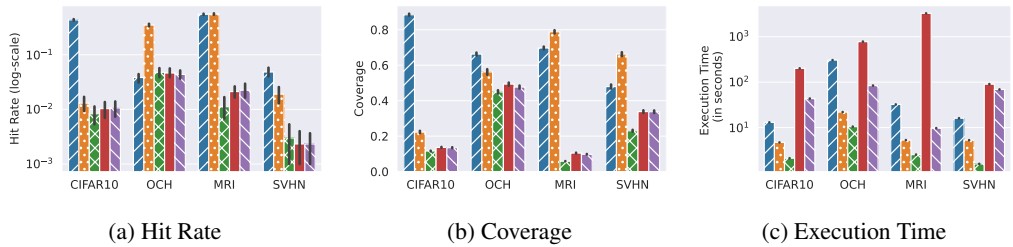

(a) Hit Rate        (b) Coverage        (c) Execution Time

Figure 6: Quantitative explanation comparison among candidate example-based explanation methods. Data augmentation strategy used is Horizontal Flip. Error bar shows 95% confidence interval. We reuse the legend of Figure 5.

frogs in Figure 3 (a) and deer on the grass in Figure 3 (b). In contrast, for the misclassified prediction case (as shown in Figure 3 (c)), we note the HD-Explain produces an example that does not even belong to the same class as the predicted one. pecifically, the predicted class is cat (while the ground truth label is bird), and HD-Explain generates an explanation sample from the deer class. This reflects a low confidence in model's prediction for the particular test example and highlight a potential error in the prediction. RPS also shows such inconsistency in explanation, which aligns with its claim (Yeh et al., 2018). The other two baseline methods do not offer such properties and still produce explanations that match the predicted label well. It is hard to justify how those training samples support such prediction visually (since no clear shared pattern is obvious to us). In addition, it is interesting to see that Influence Function and TracIn produce near identical explanations, reflecting their similarity in leveraging the perturbation of model parameters.

Figure 4 provides additional insights on SVHN dataset. HD-Explain again shows a better explanation for producing training samples that appear similar to the test samples. In addition, we notice that RPS produces the same set of explanations for different sample cases, as shown in Fig. 4 (a-b), which reveals its limitation in providing instance-level explanations. To verify this observation further, we conducted a quantitative evaluation as described in the next section.

The qualitative evaluation for OCH and MRI datasets are given in the appendix due to the page limit of the main paper. The overall observations remain consistent with CIFAR10 and SVHN.

## 4.2 QUANTITATIVE EVALUATION

In order to perform quantitative evaluation, we limit our experiments to datasets where ground-truth explanation samples are available. Specifically, given a training data sample $(\mathbf{x}_i, y_i)$, we generate a test point $\mathbf{x}_t$ by adopting two image data augmentation methods:

- **Noise Injection:** $\mathbf{x}_t = \mathbf{x}_i + \epsilon$   s.t.   $\epsilon \sim \mathcal{N}(0, 0.01\sigma)$, where $\sigma$ is the element-wise standard deviation of features in the entire training dataset.

- **Horizontal Flip:** $\mathbf{x}_t = \text{flip}(\mathbf{x}_i)$, where we flip images horizontally that do not compromise the semantic meaning of images.

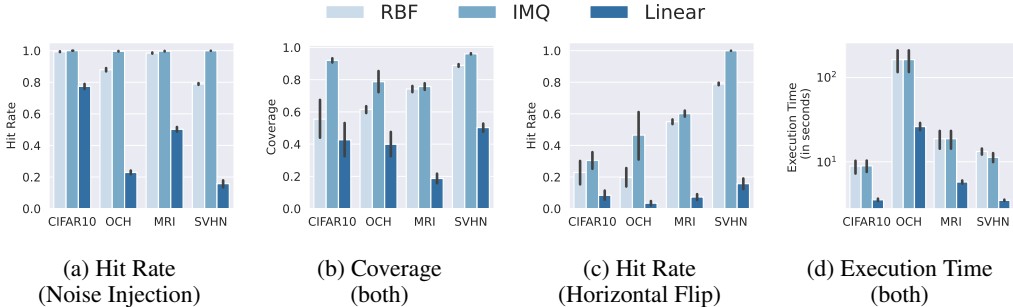

Figure 7: Quantitative explanation comparison among HD-Explainers with different kernel functions on all image classification datasets. Error bar shows 95% confidence interval.

We created 30 augmented test points for each training data point ($> 10,000$ data points) in each dataset, resulting in more than $300,000$ independent runs. Since the data augmentation is guaranteed to maintain prediction consistency, the ideally best explanation for the generated test point is the original data point $\mathbf{x}_i$ itself. Hence, the quantitative evaluation could be a sample retrieval evaluation where **Hit Rate** measures the probability of successful retrieval.

Figure 5(a) shows the hit rate comparison among candidate methods on the four image classification datasets under Noise Injection data augmentation. The existing methods face significant difficulty in retrieving the ideal explanatory sample ($\leq 10\%$), even with such a simple problem setup; only HD-Explain (and its variant) produces a reasonable successful rate ($> 80\%$). We further investigate the diversity of explanations across a testset using the **Coverage** metric. Here, diversity indirectly reflects the granularity of an explanation when accumulated over the test dataset. Figure 5(b) shows the Coverage score, the ratio of explanation samples that are unique over many test points. It turns out that existing solutions produce only 10% - 50% coverage – many test points receive the same set of explanations, disregarding their unique characteristic. We further observed that the explanations of baselines are often dominated by the class labels; data points predicted as the same class would receive a similar set of explanations. In contrast, HD-Explain shows substantially higher coverage, generating explanations that considers the unique characteristics of each test point.

Regarding computation efficiency, while we have summarized the scalability limitation of the candidate methods in Table 1, there was no computational efficiency evaluation conducted in previous works. We recorded the wall clock execution time of each experiments as shown in Figure 5(c). As expected, the Influence Function takes longer to return its explanation than other methods. HD-Explain*, TracIn* and RPS, all use the last layer to generate explanation. RPS showed the lowest compute time since it does not require auto-differentiation for computing the training data influence. HD-Explain* showed the second best compute time and is efficient than TracIn*[2] and IF. HD-Explain considers the whole model for explanation and its compute time is not directly comparable to others. However, it shows better efficiency than IF across all datasets and is better than TracIn* on CIFAR10.

We observe a similar trend in the other data augmentation scenario, Horizontal Flip, where computation time and coverage are roughly the same, as shown in Figure 6. However, we do notice that, as the outcome of image flipping, the raw feature (pixel) level similarity between $\mathbf{x}_t$ and $\mathbf{x}_i$ is destroyed. As an outcome, the HD-Explain that works on raw features suffers from performance deduction while other methods, including HD-Explain*, are less impacted. This observation suggests that choosing the layer of explanation might be considered in the practical usage of this approach.

## 4.3 KERNEL OPTIONS

We use the Radial Basis Function (RBF) as our default choice of kernel. However, another kernel may better fit a particular application domain. In this experiment, we compare three well-known kernels i.e. Linear, RBF, and Inverse Multi-Quadric (IMQ) on the three image classification datasets.

---

[2]TracIn* is configured only to compute the gradient of the prediction layer due to its high memory requirements.

Figure 7 presents the results under both data augmentation scenarios. Overall, the IMQ kernel performs better than the RBF kernel regarding explanation quality (Hit Rate). The advantage is significant when the data augmentation scenario is Horizontal Flip (Figure 7c, which appears more challenging than Noise Injection. IMQ also showed better performance on Coverage (Figure 7b). The linear kernel performs worse compared to other kernels. However, it is substantially efficient than the others, as shown in Figure 7d, highlighting its utility on large datasets. Compared to the baselines presented in Figure 5, we note that the Linear kernel is sufficient for HD-Explain to stand out from other methods in both performance and efficiency.

## 4.4 DISCUSSION: INTUITION ON WHY HD-EXPLAIN WORKS

After showing HD-Explain's empirical performance, we now present our understanding of how HD-Explain finds the explanations for the prediction of a test data point. In particular, we want to understand why the approach is faithful to the pre-trained model.

In HD-Explain, the key metric on measuring the predictive supports of a test point $\mathbf{x}_t$ given a training data $(\mathbf{x}_i, \mathbf{y}_i)$ is the KSD defined kernel $\kappa_\theta([\mathbf{x}_t||\hat{\mathbf{y}}_t], [\mathbf{x}_i||\mathbf{y}_i])$, where $\hat{\mathbf{y}}_t$ denotes the predicted class label by model $f_\theta$ in one-hot encoding. By definition, the kernel $\kappa_\theta((\mathbf{x}_a, y_a), (\mathbf{x}_b, y_b)) = k_\theta(a, b)$ between two data points can be decomposed into four terms

$$\underbrace{\text{trace}(\nabla_a \nabla_b k(a, b))}_{①} + \underbrace{k(a, b) \nabla_a \log P_\theta(a)^\top \nabla_b \log P_\theta(b)}_{②}$$
$$+ \underbrace{\nabla_a k(a, b)^\top \nabla_b \log P_\theta(b)}_{③} + \underbrace{\nabla_b k(a, b)^\top \nabla_a \log P_\theta(a)}_{④}.$$

We examine the effect of each term as follows:

- ①: We note that the first term is often a similarity bias of raw data points given a specified kernel function. In particular, for the RBF kernel $k(a, b) = \exp(-\gamma||a - b||^2)$, the first term is simply $\sum_i^{d+l} 2\gamma k(a, b)$, where $d + l$ refers to the sum of input and output (in one-hot) dimensions of a data point. Intuitively, the term shows how similar the two data points are given the RBF kernel. For linear kernel $k(a, b) = a^\top b$, on another hand, the first term is simply $d + l$ as a constant bias term, which does not deliver any similarity information between the two data points.

- ②: The second term reflects the similarity between two data points in the context of the trained model. In particular, considering the sub-term $\nabla_a \log P_\theta(a)^\top \nabla_b \log P_\theta(b)$, based on our derivation in Equation 2 (in the main paper), we note it is equivalent to

$$[\nabla_{\mathbf{x}_a} \log f_\theta(\mathbf{x}_a)_{y_a} || \log f_\theta(\mathbf{x}_a)]^\top [\nabla_{\mathbf{x}_b} \log f_\theta(\mathbf{x}_b)_{y_b} || \log f_\theta(\mathbf{x}_b)]$$
$$= \underbrace{\nabla_{\mathbf{x}_a} \log f_\theta(\mathbf{x}_a)_{y_a}^\top \nabla_{\mathbf{x}_b} \log f_\theta(\mathbf{x}_b)_{y_b}}_{\text{similarity of scores (input gradients)}} + \underbrace{\log f_\theta(\mathbf{x}_a)^\top \log f_\theta(\mathbf{x}_b)}_{\text{similarity of predictions}},$$

where both terms could be viewed as similarity between data points in the context of trained model.

- ③-④: Both of the last two terms examine the alignment between the score of one data point and the kernel derivative of another data point. We conjecture that this alignment reflects how a test prediction would change if there is a training data point present closer to it than before.

## 5 CONCLUSION

This paper presents HD-Explain, a Kernel Stein Discrepancy-driven example-based prediction explanation method. We performed comprehensive qualitative and quantitative evaluation comparing three baseline explanation methods using three datasets. The results demonstrated the efficacy of HD-Explain in generating explanations that are accurate and effective in terms of their granularity level. In addition, compared to other methods, HD-Explain is flexible to apply on any layer of interest and can be used to analyze the evolution of a prediction across layers. HD-Explain serves as an important contribution towards improving the transparency of machine learning models.

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

## A    APPENDIX / SUPPLEMENTAL MATERIAL

## B    PROBLEM DEFINITION RECAP

We consider the task of explaining the prediction of a differentiable classifier $f : \mathbb{R}^d \rightarrow \mathbb{R}^l$, given inputs test sample $\mathbf{x}_t \in \mathbb{R}^d$, where $d$ denotes the input dimension and $l$ denotes the number of classes. Specifically, we are interested in explaining a prediction of a model $f(\cdot)$ by returning a subset of its training samples $D = \{(\mathbf{x}_i, y_i)\}_{i=1}^n$ that has strong predictive support to the prediction of test point $\mathbf{x}_t$. While not explicitly stated in the main paper, we treat example-based prediction explanation as a function $\psi(f, D, \mathbf{x}_t) : \mathcal{F} \times \mathcal{D} \times \mathbb{R}^d \rightarrow \{\mathbb{R}^d, \mathbb{R}^l\}^k$ such that it takes a trained model $f$, a training dataset $D$, and an arbitrary test point $\mathbf{x}_t$ as inputs and output top-$k$ training samples as explanations.

## C    ADDITIONAL DERIVATION OF KERNELIZED STEIN DISCREPANCY

While Stein's Identity has been well described in many previous works (Liu et al., 2016; Liu & Wang, 2016; Chwialkowski et al., 2016), we briefly recap some key derivations in this paper to seek for self-contained.

As mentioned in the main paper, Stein's Identity states that, if a smooth distribution $p(x)$ and a function $\phi(x)$ satisfy $\lim_{||x|| \rightarrow \infty} p(x)\phi(x) = 0$, we have

$$\mathbb{E}_{x \sim p}[\phi(x)\nabla_x \log p(x) + \nabla_x \phi(x)] = \mathbb{E}_{x \sim p}[\mathcal{A}_p \phi(x)] = 0, \quad \forall \phi.$$

Intuitively, by using integration by part rules, we can reveal the original assumption from the derived expression such that

$$\int_x \phi(x)\nabla_x \log p(x) + \nabla_x \phi(x)dx = p(x)\phi(x)\Big|_{-\infty}^{+\infty}$$

Stein Discrepancy measures the difference between two distributions $q$ and $p$ by replacing the expectation of distribution $p$ term in Stein's Identity expression with distribution $q$, which reveals the difference between two distributions by projecting their score functions (gradients) with the function $\phi(x)$

$$\max_{\phi \in \mathcal{F}} \mathbb{E}_{x \sim q}[\mathcal{A}_p \phi(x)] = \max_{\phi \in \mathcal{F}} \mathbb{E}_{x \sim q}[\mathcal{A}_p \phi(x)] - \underbrace{\mathbb{E}_{x \sim q}[\mathcal{A}_q \phi(x)]}_{=0}$$

$$= \max_{\phi \in \mathcal{F}} \mathbb{E}_{x \sim q}[\underbrace{\phi(x)}_{\text{projection coefficients}} \underbrace{(\nabla_x \log p(x) - \nabla_x \log q(x))}_{\text{score function difference}}]$$

Clearly, the choice of projection coefficients (function $\phi(x)$) term is critical to measure the distribution difference.

Kernelized Stein Discrepancy (KSD) addresses the task of searching function $\phi$ by treating the above challenge as an optimization task where it decomposes the target function $\phi$ with linear decomposition such that

$$\max_{\phi \in \mathcal{F}} \mathbb{E}_{x \sim q}[\mathcal{A}_p \phi(x)] = \max_{\phi \in \mathcal{F}} \mathbb{E}_{x \sim q}[\mathcal{A}_p \sum_i w_i \phi_i(x)] = \max_{\phi \in \mathcal{F}} \sum_i w_i \mathbb{E}_{x \sim q}[\mathcal{A}_p \phi_i(x)],$$

with linear property of Stein operator $\mathcal{A}_p$. The linear decomposition path is the way to reduce the optimization task into looking for a finite number of the base functions $\phi_i \in \mathcal{F}$ whose coefficient norm is constraint to 1 ($||\mathbf{w}||_{\mathcal{H}} \leq 1$). KSD takes $\mathcal{F}$ to be the unit ball of a reproducing kernel Hilbert space (RKHS) and leverages its reproducing property such that $\phi(x) = \langle \phi(\cdot), k(x, \cdot) \rangle$, which in turn transforms the maximization objective of the Stein Discrepancy into

$$\max_\phi \langle \phi(\cdot), \mathbb{E}_{x \sim q}[\mathcal{A}_p k(\cdot, x)] \rangle_{\mathcal{H}}, \quad s.t. ||\phi||_{\mathcal{H}} \leq 1.$$

The optimal $\phi$ is therefore a normalized version of $\mathbb{E}_{x \sim q}[\mathcal{A}_p k(\cdot, x)]$. Hence, KSD is defined as the optimal between the distribution $p$ and $q$ with the optimal solution of $\phi$

$$\mathbb{S}(q, p) = \mathbb{E}_{x, x' \sim q}[\kappa_p(x, x')], \quad \text{where} \quad \kappa_p(x, x') = \mathcal{A}_p^x \mathcal{A}_p^{x'} k(x, x').$$

## D    DISCUSSION: RELAXATIONS OF KSD ESTIMATION

In section 3.1, we introduced multiple relaxations such that KSD estimation can support predictive model (as conditional distribution model) with discrete labels. However, we want to clarify that the relaxations introduced are not generally applicable to other context (e.g., Goodness of Fit) given their selective conditions on the data input distribution. In fact, the data distribution $P$ that generated the datasets $\{(\mathbf{x}_i, y_i)\}_i^n$ could be complex given the potentially intractable marginal distribution of $P(\mathbf{x})$. Our solution avoids the modelling of such complexity by only limiting the KSD discrepancy computation between model distribution $P_\theta$ and the sampled data distribution $P_D$ instead of touch original distribution $P$.

In particular, in the data-centric prediction explanation context, we only aim to extract training data points that are similar to the test sample, such that all inputs in the framework are valid (e.g., images) rather than random continuous inputs sampled from an unknown distribution $P$ that might be inevitable in other contexts. To apply the similar relaxation to other context, a thorough theoretical proof is needed, which is out of the scope of this research.

It is worth noting that the both relaxations introduced in this paper has corresponding theoretically rigid solutions (Yang et al., 2018; Jitkrittum et al., 2020) with the cost of computational overhead. While they are much more elegant, for explanation prediction purpose, we may prefer faster approximations. How to better incorporate the theoretically rigid solution in a more efficient way will be in our future research.

## E    DISCUSSION: PREDICTION EXPLAIN QUALITY ON HEALTHCARE DATASET

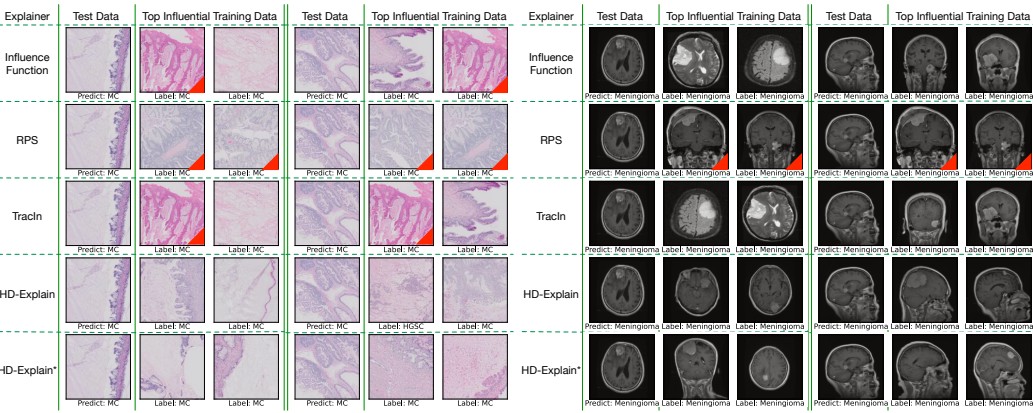

(a) Overian Cancer Histopathology (OCH)                    (b) Brain Tumor MRI

Figure 8: Qualitative evaluation of example-based explanation methods on Overian Cancer histopathology and Brain Tumor MRI datasets. We show two test data points that are predicted to belong to the same class in each dataset. Red triangle in the top right corner of an image shows the duplicate explanations across test samples.

Figure 8 provides additional insights into Ovarian Cancer histopathology and Brain Tumor MRI datasets. HD-Explain again shows a better explanation for producing training samples that appear similar to the test samples (note for the semantic similarity, these explanations should be referred to a medical practitioner). For instance, the explanation of HD-Explain follows the scanning orientation of test points in MRI as shown in Figure 8 (b). We note all baseline approaches tend to produce similar explanations to test samples belonging to the same classes. Rather than providing individual prediction explanations, those approaches act closer to per-class interpreters that look for class prototypes. To verify this observation further, we conducted a quantitative evaluation as described in the next section.

There is potential concern on interpreting our experiments on the two healthcare datasets provided, where no domain experts' evaluation contained in this work.

In the quantitative evaluation (Figures 5 and 6), we highlight that HD-Explain demonstrates better explanation performance in terms of retrieving original data points that were used for data augmentation-based retrieval tests. This study is objective, requiring no domain knowledge for result validation.

Regarding the qualitative evaluation (as part of the visualization in Figure 4), we agree that the explanation quality of healthcare datasets needs insights from domain experts, whereas machine learning research in general often lacks corresponding authority to justify specific domain's results. This challenge extends across the entire model explanation literature. Our objective is to facilitate the general population's understanding, even in the absence of domain expert evaluation. It is important to recap that the provided explanations by HD-Explain are visually similar to the test data points. However, their deeper pathological interpretation requires further investigation by healthcare practitioners, which we encourage for future studies. Furthermore, experimental evaluation of our approach on CIFAR demonstrates that the effectiveness of our method is not limited to medical datasets and can be easily applied across other domains. Overall, we believe that qualitative evaluation aims to enhance understanding of the model's behavior rather than serving as a performance justification.

## F    Dataset Details

Table 2: Summary of datasets used in the paper.

| Dataset | Application | Type | # Size | # Feature Dimension | # Number of Classes | Duplicated Samples | Public Dataset |
|---------|-------------|------|--------|---------------------|---------------------|--------------------|----------------|
| Two Moons | Synthetic | 2D Numeric | 500 | 2 | 2 | No | Shared with code |
| Rectangulars | Synthetic | 2D Numeric | 500 | 2 | 3 | No | Shared with code |
| CIFAR-10 | Classification Benchmark | Image | 60,000 | $32 \times 32 \times 3$ | 10 | No | Yes |
| Overian Cancer | Histopathology (**Private**) | Image | 20,000 | $128 \times 128 \times 3$ | 5 | Yes | No |
| Brain Tumor | MRI Benchmark | Image | 7,023 | $128 \times 128 \times 3$ | 4 | Yes | Yes |

In this paper, we conducted our experiments on five datasets – two synthetic and three benchmark image classification datasets. As the work concerns the trustworthiness of the machine learning model in high-stakes applications, we also introduced medical diagnosis datasets to provide more insight into the potential benefit the proposed work introduced. To train the target machine learning models, we conducted data augmentations to increase the number of training data samples, including random cropping, rotation, shifting, horizontal flipping, and noise injection. Table 2 summarizes more details about the datasets.

## G    Data Debugging

Before describing the data debugging setting of this paper, we want to recap that the data debugging functionality is a side effect/benefit of HD-Explain, which is not our main proposal. Indeed, using the prediction explanation method as a data debugging tool is still under investigation since it might be over-claimed due to the over-regularized setting in previous works (e.g. Binary classification tasks). While we relaxed some settings, we don't claim it practical for real-world applications.

The data debugging task in this paper is a data sample retrieval task where we retrieve samples that intentionally flipped their classification labels. Higher Precision and Recall of the retrieval reflects higher performance of data debugging.

For the HD-Explain (and its variant HD-Explain*), the retrivial order is determined by the values of the diagonals of the KSD-defined kernel matrix, $\kappa_\theta(a, a)$ for all $a \in D$. This setting is very similar to how the Influence function does the data debugging with the self-influence of a data sample. Indeed, $\kappa_\theta(a, a)$ could be treated as a self-influence that does not rely on model parameters.

Now, we describe our data debugging experiments to highlight the self-explanatory ability of candidate methods on the training data. In particular, we generalized previous research's binary classification-based data debugging experiment into a multi-classification scenario, where we randomly flip labels of 100 training data points at each run. We adopt standard information retrieval metrics, Precision and Recall, that measure how likely the candidate methods can retrieve the mislabeled training data points. Figure 9 shows our experimental results. While HD-Explain on the entire model has little

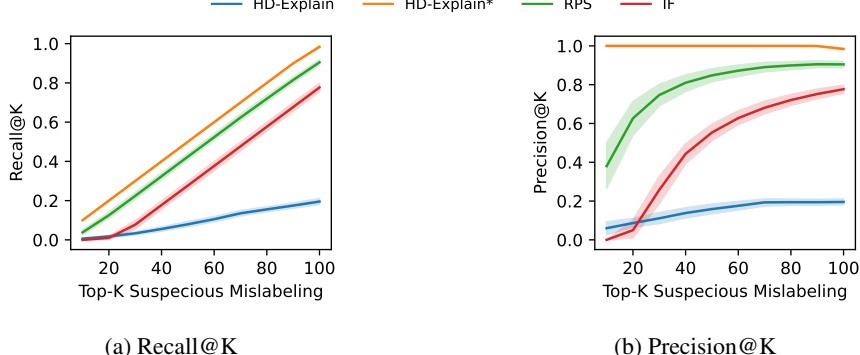

Figure 9: Data debugging comparison among candidate methods on CIFAR-10 dataset. Results collected from 30 independent runs. Error bar shows 95% confidence interval.

data debugging ability, its variant on the last layer offers outstanding performance compared to the other last-layer explanation methods. Note, the data debugging functionality is a side effect/benefit of HD-Explain, which is not our main proposal

## H    HARDWARE SETUP

We ran all our experiments on a machine equipped with a GTX 1080 Ti GPU, a second-generation Ryzen 5 processor, and 32 GB of memory.

## I    BROADER IMPACT

The development of HD-Explain, a highly precise and data-centric explanation method for neural classifiers, promises to significantly enhance the transparency and trustworthiness of machine learning models across various applications. Furthermore, HD-Explain's scalable and computationally efficient approach makes it feasible for deployment in large-scale, real-world applications. This not only promotes transparency and accountability in AI systems but also paves the way for broader acceptance and integration of AI technologies in society. By bridging the gap between complex model behavior and human understanding, HD-Explain fosters a more informed and trust-based relationship between AI systems and their users. Overall, HD-Explain's contributions to model interpretability and transparency have the potential to drive significant advancements in the responsible and ethical use of AI, ensuring that these technologies are developed and deployed in ways that are understandable, accountable, and aligned with societal values.

However, in terms of **Negative Societal Impacts**, over-reliance on explanation methods like HD-Explain might create a false sense of interpretability, masking the inherent limitations and uncertainties of machine learning models. Thus, careful consideration and mitigation of these negative impacts are crucial for the responsible deployment of HD-Explain and similar tools.

## J    LIMITATION OF PROPOSED METHOD

While HD-Explain demonstrates significant promise in providing detailed explanations for neural classifier predictions, we intended to investigate the limitation of our method.

For a given datapoint, we sorted all data based on their Stein Kernel similarity to the target data and found that the top relevant data points selected by HD-Explain were very similar in attributes such as colour palette, object position, and background colour, which are unique to the target sample (See Figure 10 Left). This observation triggered our curiosity about whether the HD-Explain is sensitive to its raw input features (low-level information). To investigate this possibility, we set a low threshold to capture a large portion of relevant data points. We observed that for data points with low Stein Kernel similarity (i.e. they are weakly relevant), the dot product of Stein scores is low but still above this datapoint designated threshold. We noticed that the model's prediction confidence is lower for such outliers due to the reduced dot product of Stein scores, indicating a reliance on RBF similarity.

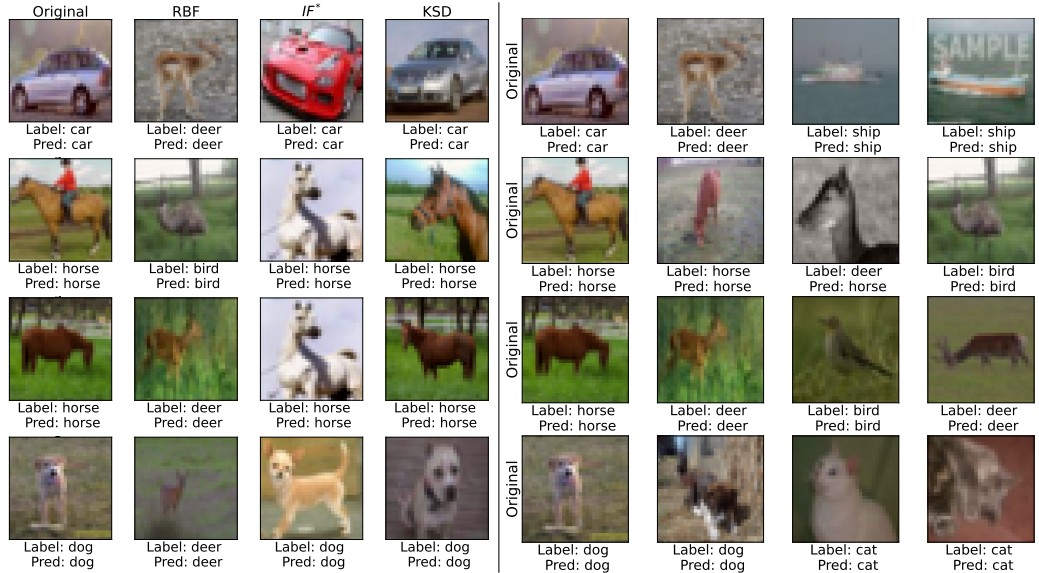

Figure 10: Demonstrative figures to show the limitations of our method on CIFAR-10 dataset. Left) Given query sample, the most relevant data points selected by different methods, such as RBF kernel similarity (RBF), influence-based methods (IF*), and HD-Explain. Right): By setting a low threshold on the KSD, we note HD-Explain can produce less informative explanation. HD-Explain still falsefully considered them as explanation. Our conjecture is that it relies more on the RBF similarity.

Such that, HD-Explain's performance depends on the model's prediction quality and generalization ability. As HD-Explain incorporates the complete derivative calculation, it includes more information from the model's internal weights, but generalization errors, such as overlapping decision boundaries and the impact of noisy data, can cause the selection of data points from other classes, especially for correctly labelled but low-confidence predictions. To balance the potential drawback of modelling the raw feature space, the user can use HD-Explain*.

In the main paper, we evaluated HD-Explain with three well-known kernel options, including Linear, RBF, and Inverse Multi-Quadric (IMQ). The purpose was to demonstrate the impact of kernel choices on the performance of HD-Explain. While we show that IMQ performs best in our experimental environment, we want to highlight that the selection of the kernel may influence the HD-Explain's performance in practice. Hence, one may need to conduct empirical analysis on which kernel to use before applying HD-Explain in production.

## K    RELATION TO DATA ATTRIBUTION ESTIMATION

Data attribution estimation is closely related to the sample-based prediction explanation but a different concept. As stated by Park et al. (2023b), the definition of data attribution is as follows:

**Definition K.1** (*Data attribution*). Consider an ordered training set of examples $S = \{z_1, \ldots, z_n\}$ and a model output function $f(z; \theta)$. A *data attribution method* $\tau(z, S)$ is a function $\tau : \mathcal{Z} \times \mathcal{Z}^n \to \mathbb{R}^n$ that, for any example $z \in \mathcal{Z}$ and a training set $S$, assigns a (real-valued) score to each training input $z_i \in S$ indicating its importance to the model output $f(z; \theta^*(S))$. When the second argument $S$ is clear from the context, we will omit the second argument and simply write $\tau(z)$.

In particular, data attribution estimation faces a more generalized problem that does not explain the importance of data for a specific pre-trained model but the importance of a family of models of the same architecture or function. The star ($\theta^*$) refers to the potentially optimal model that can be trained on the dataset $S$. Indeed, if we examine the two approaches in the data attribution estimation literature (data modelling (Ilyas et al., 2022) and TRAK (Park et al., 2023b)), we note both approaches require either training multiple models on a subset of data points or introducing various aggressive approximations, such as (1) linear Taylor expansion, 2) random projection, and 3) newton approximation. Both data and model manipulations will cause unfaithfulness to the pre-trained model.

In fact, the data modelling algorithm (Ilyas et al., 2022) does not involve any pre-trained model but directly optimizes the linear data modelling score as follows.

$$\tau_{\text{DM}}(z) := \min_{\beta \in \mathbb{R}^n} \frac{1}{m} \sum_{i=1}^{m} (\beta^\top 1_{S_i} - f(z; \theta^*(S_i)))^2 + \lambda \|\beta\|_1.$$

## L    HD-EXPLAIN: EXPLANATION PROCESS

The following algorithm shows the algorithm of HD-Explain in pseudocode.

---
**Algorithm 1** HD-Explain
---
**Input:** Training set $D$, Test input $\mathbf{x}_{\text{test}}$, and classifier model $f_\theta$
**Output:** Sample based explanations $D_{\text{explain}} \subset D$

1: **Step 1: Caching**                                                             ▷ Reduce redundant computation
2: initialize empty list $c \leftarrow []$
3: **for** $(\mathbf{x}_i, y_i) \in D$ **do**
4:     $\mathbf{p}_i \leftarrow f_\theta(\mathbf{x}_i)$
5:     $\mathbf{g}_i \leftarrow \nabla_{\mathbf{x}_i} \log f_\theta(\mathbf{x}_i)_{y_i}$
6:     c.add($[\mathbf{x}_i, y_i, \mathbf{p}_i, \mathbf{g}_i]$)
7: **end for**

8: **Step 2: Prediction Contribution of Each Training Data**
9: Given test input $\mathbf{x}_{\text{test}}$
10: $\mathbf{p}_{\text{test}} \leftarrow f_\theta(\mathbf{x}_{\text{test}})$
11: $\hat{y}_{\text{test}} \leftarrow \text{argmax } \mathbf{p}_{\text{test}}$                    ▷ Best Predicted Label
12: $\mathbf{g}_{\text{test}} \leftarrow \nabla_{\mathbf{x}_{\text{test}}} \log f_\theta(\mathbf{x}_{\text{test}})_{\hat{y}_{\text{test}}}$
13: c.add($[\mathbf{x}_{\text{test}}, y_{\text{test}}, \mathbf{p}_{\text{test}}, \mathbf{g}_{\text{test}}]$) $\nabla_{\mathbf{x}_{\text{test}}, \hat{y}_{\text{test}}} \log P_\theta(\mathbf{x}_i, \hat{\mathbf{y}}_i) \leftarrow [\mathbf{g}_{\text{test}} || \mathbf{p}_{\text{test}}]$
14: **for** $(\mathbf{x}_i, y_i) \in D$ **do**
15:     $\nabla_{\mathbf{x}_i, \mathbf{y}_i} \log P_\theta(\mathbf{x}_i, \mathbf{y}_i) \leftarrow [\mathbf{g}_i || \mathbf{p}_i]$      ▷ Cache-able if needed
16:     $\kappa_\theta((\mathbf{x}_i, y_i), (\mathbf{x}_{\text{test}}, \hat{y}_{\text{test}})) \leftarrow$ Equation 1
17: **end for**
18: $D_{\text{explain}} \leftarrow \text{argsort}([\kappa_\theta((\mathbf{x}_i, y_i), (\mathbf{x}_{\text{test}}, \hat{y}_{\text{test}}))$ for $i \in |D|])$

---

## M    ADDITIONAL QUALITATIVE EVALUATION EXAMPLES FOR SVHN

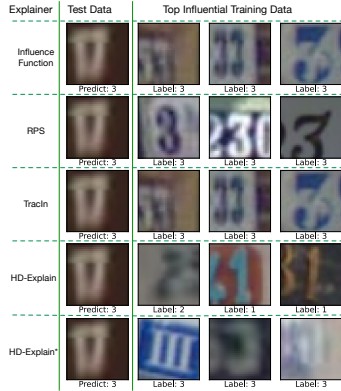

(a) Low-Conf Incorrect Predict

Figure 11: Qualitative evaluation of various example-based explanation methods using SVHN. We show the missing scenario in the main paper, where the target model makes low-confident prediction that does not match ground truth label (which is a 7). For the sub plot, we show top-3 influential training data points picked by the explanation methods for the test example. The observation is similar to that of CIFAR-10.

