# OpenReview forum: "Data-centric Prediction Explanation via Kernelized Stein Discrepancy"
_ICLR.cc/2025/Conference — ICLR 2025 Poster_

### Official Review · Reviewer_o2SA · 2024-10-29

**Soundness:** 2
**Presentation:** 1
**Contribution:** 3
**Rating:** 6
**Confidence:** 3

**Summary:**

The paper proposes a new example-based explanation technique by using Kernelized Stein Divergence (KSD) to identify coresets of training samples that best provide support for a particular prediction on a test point by the model.

**Strengths:**

The biggest strength of the paper is the simple novelty of the idea. Coresets have a established history already and using a similar approach towards finding points in the training set that explain the model prediction for a test point is a great idea that seems to have not been tried before.

**Weaknesses:**

The biggest weakness of the paper is poor presentation and writing. The paper is unfortunately littered with typos, grammatical mistakes, and run-on sentences which make it sometimes very difficult to understand certain passages.

The other weakness is the experiments section, especially the qualitative experiments in Figures 3 and 4 that do not seem to clearly portray the proposed method's strength when compared to the competitors and it is left to the reader to figure it out.

here's a few typos and possible presentation improvements:

line 93-94: currently it reads like the research history spans from 2020 to 2020. Please add that the papers cited are survey papers.

line 106-107: "unfortunately, those approaches are often with strong assumption of good prototypes (as of training data distribution)", this sentence is not self explanatory. Please expand on what "good" prototypes means here and what strong assumptions.

line 138: to assess the goodness-of-fit ... of what?

Figure 1: y-axis should say $P_\theta$ in markdown

Figure 2 will probably make a better Figure 1 - it shows why the proposed method could potentially work better.

line 235: "that out of" -> "that is out of"

line 237-240: run-on sentence and ends abruptly without a conclusion.

line 319: "high-confident" -> "high-confidence"

line 323: "we note the HD-Explain produces an example that does not even..." in figure 3(c) all examples are labelled as cat, so not sure which example this sentence is referring to, and it's visually very hard to distinguish the images.

line 366: "target model makes 1-2)" -> remove "-2"

line 428: refers to Figure 8 and conclusions drawn from it as "previous observation". There is no Figure 8 in the main manuscript. Figure 8 is in the appendix and shows up later.

Conclusions section is inadequate and should include discussion on limitations and future directions. ICLR allows for 10 pages of text this year.

**Questions:**

Can the authors expand on the use of kernels. In my experience euclidean distance based kernels (like RBF kernel used in the paper) tend to fall apart when used with high dimensional data - like images, due to curse of dimensionality. Given the reliance of this method on being able to compute useful kernels how do the authors think this method can be scaled to actual images (that are significantly higher dimensional than the benchmark images used here)?

In Figure 4, the authors say the "model predictive accuracy is near 100%, ... random search". Why do a random search here? Shouldn't you be able to pull any mispredicted samples deterministically?

In the quantitative eval section why is only horizontal flip tested? Would other transformations of data points hold just as well? (e.g. rotation -, Figure 1 seems to suggest that if the whole dataset is rotated that could impact KSD)

---

> ### Author Response · Authors · 2024-11-19
>
> Thank you very much for your feedback. Below is our response to the concerns and questions you raised.
>
> #### **Moving content from appendix to main paper:**
> We appreciate the reviewer pointing out the page limitation issue. Indeed, we did not notice the page limit change in the paper call. **We have now updated our paper draft by moving some sections into the main paper as suggested by the reviewer**.  Now the main paper is 10 pages.
>
> #### **Correcting typographical errors:**
>
> We greatly appreciate the time and effort you have dedicated to reviewing our paper. In the revised version, we have carefully addressed the suggested improvements.
>
> With regard to the concept of **goodness of fit**, we would like to clarify that it is not a typo. This term is widely used in statistical analysis to evaluate how well a set of observed values aligns with the expected values predicted by a given model.
>
> #### **Qualitative evaluation:**
>
> Including qualitative evaluation by showing image examples is quite common in the sample-based explanation research field [1]. We followed the convention by introducing the corresponding observations to show the difference. **We agree that concluding qualitative evaluation alone is somewhat subjective, which motivated us to include further quantitative evaluation that was not included in previous works.** However, the qualitative evaluation in our experiments does show weaknesses of existing solutions (e.g., producing the same explanations for different test cases). This observation, at least, shows that the existing approaches produce explanations for a class more than that of the specific test sample, which HD-Explain does not suffer from.
>
> #### **Kernel Evaluation:**
>
> We exploited multiple commonly used kernel options and showed they all perform well in practice. We are happy to try more, but we note this approach is proposed to introduce the Kernelized Stein Discrepancy for the post-explanation of models and serve as the baseline for future research. Our proposed method can support various kernels that people may want to choose in practice. Introducing new kernels in the paper won't impact the claim of this work
>
> #### **Predictive confidence:**
>
> To ensure the robustness of experimental evaluation and to eliminate the possibility of cherry-picking test samples, in our experiment setup, we always conduct random sampling on the test dataset and evaluate the performance of our model on these subsets. Across all sampled subsets, we indeed didn't find any wrong prediction.
>
> #### **Experimental evaluation validity:**
>
> To avoid the generation of invalid samples, and keep the validity of manipulated samples we avoided rotation-vertical flipping.  In some cases such as number “6” vertical flipping results in number “9”.
>
>
> [1] Koh, Pang Wei, and Percy Liang. "Understanding black-box predictions via influence functions." International conference on machine learning. PMLR, 2017.

---

> > ### Comment · Reviewer_o2SA · 2024-11-19
> > **Response**
> >
> > **Correcting typographical errors**: Thank you for fixing some of the typographical errors. There is still presentation issues that remain. e.g. Figure captions (3 and 4) have figures labeled as a, b, c but the caption refers to the figures as 1, 2, 3... please consider doing a thorough read-through of the paper to catch these.
> >
> > regarding 'goodness-of-fit' I understand what it is, you still want to finish the sentence e.g. goodness-of-fit of the model.
> >
> > **Qualitative evaluation**: adding the true label in that figure seems to help assuage some of my confusion. It's still relatively hard to see the improvement qualitatively, but I agree it's always subjective.
> >
> > **Kernel Evaluation**: I agree that finding a better kernel doesn't need to be a thrust of this paper. However this is a limitation of this method and should find a place in conclusions.
> >
> > **Predictive Confidence**: It makes sense not to cherry-pick. However if "Across all sampled subsets, we indeed didn't find any wrong prediction" there is nothing wrong in doing a deterministic check for if the there was any wrong prediction at all. And if there was, analyze that. Otherwise the impression is that the method didn't get anything wrong. which may or may not be true.

---

> > > ### Author Response · Authors · 2024-11-24
> > >
> > > Thank you for your valuable suggestions and for recognizing our efforts in addressing your earlier comments.
> > >
> > > **Typographical error correction:**
> > >
> > > Thank you for providing further suggestion. **We have updated our paper draft by correcting typos we can observe, including the paper caption problem you pointed out.**  We also made several minor edits to improve the readability of section 4.1 (e.g. recapping the ground-truth class labels of Figures 3 and 4). The corresponding updates are shown in red (mostly in Figures 3, 4 and Section 4.1).
> > >
> > > **Kernel Evaluation:**
> > >
> > > Thanks for acknowledging that kernel options are not the focus of this paper and are not the limitation of this work. Given your suggestion, **we have included one new paragraph in the revision (Appendix: Section J) to discuss the potential impact of choosing kernels.** In short, given our observation on the three well-known kernels (Linear, RBF, and IMQ) we have tested in the paper, we acknowledge that HD-Explain's performance could be impacted by the choice of kernel and we suggest the practitioners to select the kernel (through empirical analysis if needed) before using HD-Explain in practice to best fit their application domain.
> > >
> > > **Misclassified SVHN sample:**
> > >
> > > Given your suggestion, we did an exhaustive search and spotted the misclassified sample (See Appendix Section M). The observation aligns with that of the CIFAR-10 dataset. Specifically, when the classifier makes a wrong prediction, prediction explanations produced by HD-Explain likely include samples that do not even belong to the same class as the predicted one. It could be a sign of a potentially wrong prediction.
> > >
> > > While we obtained the misclassified sample for the SVHN dataset, we kept the two "High-Conf Correct Predict" in Figure 4 to show the problem of the existing solution (RPS), which tends to produce class-level explanations rather than instance-level explanations. The HD-Explain does not suffer from this explanation granularity problem.

---

> > > > ### Comment · Reviewer_o2SA · 2024-11-26
> > > > **Acknowledgement**
> > > >
> > > > Thank you for your response and for making the needed modifications in the paper. I will stay with my original score of 6. This paper does have potential to be published at ICLR.

---

### Official Review · Reviewer_3D8W · 2024-10-31

**Soundness:** 3
**Presentation:** 4
**Contribution:** 3
**Rating:** 8
**Confidence:** 3

**Summary:**

This paper proposes a new method for computing instance-level explanations, based on the Kernelized Stein Discrepancy, by poiting to examples having the biggest contribution to the classification made by the model. Indeed, given a datapoint x, the methods aims at searching the top-k training datapoints leading to the biggest value of the defined KSD kernel. The approach is tested on both classical benchmark datasets from the literature (CIFAR-10, SVHN) as well as real-life datasets needing explanations (Brain Tumor, Ovarian Cancer) and compared to a few methods from the literature.

**Strengths:**

The proposed approach is both sound and original. The method is clearly explained.

**Weaknesses:**

1. While HD-Explain* is simply presented as a declination of HD-Explain, details should be part of the work since it is equally important to HD-explain in the experiment section. In the same line of thoughts: it is said that « The qualitative evaluation for OCH and MRI datasets are given in the appendix due to the page limit of the main paper. », but the current submission makes 9 pages out of a maximum of 10 (https://iclr.cc/Conferences/2025/CallForPapers).

2. « In addition, as Influence Function and TracIn face scalability issues, we limit the influence of parameters to the last layer of the model so that they can work with models that contain a large number of parameters. Our experiments use ResNet-18 as the backbone model architecture (with around 11 million trainable parameters) for all image datasets » Since datasets such as CIFAR-10 can tackled decently by small neural networks, it would be necessary to test whether limiting the influence of parameters to the last layer of the model for benchmarks impacts the results or not. It yields a huge effect on HD-explain, as demonstrated by HD-explain*, so it might as well for other approaches.

3. I feel like there are a few conceptual incoherence/false statements in the presented approach, both theoretically and in the empirical section.

3.1 Line 320: « For both correct prediction cases, we are confident that HD-Explain provides a better explanation than others in terms of visually matching test data points e.g. brown frogs in Figure 3 (a) and deer on the grass in Figure 3 (b). » It is not because the test and the explanation images look alike to us that the model did use the information contained in the explanation to classify the test. Following the line of thought presented in the article, the explanation image should be near the test image in the feature space, yet these images might not particularly look alike to us. In short: I don’t think the observations made in section 4.1 tell us anything about the quality of the explanation.

3.2 I don’t feel like the considered metrics actually reflect the quality of the approaches. For instance, using the coverage metric requires making the assumption that a wide array of training samples should be used for explaining a wide array of test images. One could easily imagine a setup where an image contains the typical representation of a class, each other image being the same as that first one, but each having a unique particularity. In this setup, the most relevant explanation would be the typical representation for every image, since it is the most similar to every other image. When it comes to hit rate, once again, we can imagine that once a picture is flipped or noised, it now more resembles another picture.

**Typos and such**

-Figure 1, right : Error in the Y-axis name (wrong letter, d instead of D; « heta » instead of $\theta$).

**Questions:**

-Could the authors cite other works using these metrics (Hit rate, Coverage)?
-Could the authors provide explanations regarding the observations made in 4.3?
-Did the authors prevent undesired results for the image flip on SVHN? For example, flipping a 2 leads to a 5.

---

> ### Author Response · Authors · 2024-11-19
>
> Thank you very much for your feedback. Below is our response to the concerns and questions you raised.
>
> #### **Moving content from appendix to main paper:**
> We appreciate the reviewer pointed the page limitation issue out. Indeed, we did not notice the page limit change in the paper call. **We have now updated our paper draft by moving some sections into the main paper as suggested by the reviewer**.  In particular, we moved the HD-Explain* description and one discussion section (the previous Appendix D: Discussion) into the main paper to help reader to capture the intuition of this work.
>
> #### **Potential better performance of Influence Function:**
> We understand the reviewer's curiosity about seeing if we can push up the influence function's performance by letting it run on the full neural network instead of the last layer. Despite the fact that the **Influence Function in this paper was implemented exactly as it was proposed in its original paper, where it is meant to run on the last layer** (see section 5.1, page 5 of [5]), we note there is a scalability concern even if we use a smaller neural network. As shown in our paper (Table 1), the complexity of the influence function is bounded by the computation complexity of measuring the Hessian-vector product that needs multiple iterations to approximate. In addition, Resnet-18 is already a small network compared to the models that are used in real applications. Further compression may lead to practical concern even if it shows that the influence function can do better on small models.
>
> #### **Evaluation Setting:**
> Including qualitative evaluation by showing image examples is quite common in the sample-based explanation research field [5]. We followed the convention by introducing the corresponding observations to show the difference. While we agree that apparently similar images are not necessarily the best explanations, they do have positive correlations. In addition, the qualitative evaluation shows some existing solutions' weaknesses in producing the same explanations for different test cases. This observation, at least, shows that the existing approaches produce explanations for a class more than that of the specific test sample, which HD-Explain does not suffer from.
>
> As the reviewer may be concerned that the qualitative evaluation does not provide sufficient confidence in its explainability, we also included the quantitative evaluation to provide further support. Our quantitative evaluation metrics (hit rate and coverage) come from classic information retrieval literature based on the understanding that the sample-based explanation is essentially a ranking problem (where the explanation algorithm should return a ranked list of training samples as high influencers). Here, we also want to highlight that no agreed or standard quantitative evaluation metrics were proposed in previous work. The previous sample-based explanation often only includes qualitative evaluation demonstrating some side benefit of the proposed algorithms. E.g. data mislabeling debugging.
>
> We agree that the proposed evaluation setting is imperfect in capturing all possible counter-examples, but we note they have served as the best in multiple aspects. E.g. Noise Injection. While the setting looks trivial at first glance, it turns out that existing solutions (the well-known baselines) produce only 10\% - 50\% coverage -- many test points receive the same set of explanations, disregarding their unique characteristic. This interesting observation suggests the weakness of the existing approaches on such simple cases where it can happen in practice -- find typical medical exam cases with exactly the same symptoms as the new case (in a noise level difference).
>
> #### **Additional Discussion about Section 4.3 Kernel Choices:**
>
> In our experimental study, we compare the performance of three well-known kernels: Linear, RBF, and Inverse Multi-Quadric (IMQ), across three image classification datasets. Figure 7 presents the results under two data augmentation scenarios. Overall, the IMQ kernel outperforms the RBF kernel in terms of explanation quality (Hit Rate), with a notable advantage in the Horizontal Flip scenario (Figure 7.c), which appears more challenging than Noise Injection. Additionally, IMQ demonstrates superior performance in Coverage (Figure 7.b). The linear kernel, while performing worse than the other kernels in terms of explanation quality, is significantly more efficient, as illustrated in Figure 7.d. This efficiency makes it particularly useful for large datasets. Compared to the baselines in Figure 5, the Linear kernel enables HD-Explain to excel over other methods in both performance and efficiency.

---

> ### Author Response · Authors · 2024-11-19
> **Official Comment by Authors (Pt. 2)**
>
> We acknowledge that exploring additional kernel options enriched this experiment. However, our primary objective here is to highlight the importance of hyper-parameter tuning, providing practitioners with the flexibility to choose the most suitable kernel for their specific needs.
>
> In the literature [1-4], the IMQ kernel function is typically selected due to its good theoretical characteristics and practical effectiveness. As shown in [3], the IMQ kernel, together with the Stein operator provides a KSD that controls weak convergence, and the established guaranteed convergence. We didn't do extra work to validate the image after the horizontal flip. It is due to the observation that model performance can improve significantly with horizontal flip data augmenting during model training, and the model can identify numbers with a horizontal flip. Indeed, humans can identify numbers from a mirror as well. We didn't encounter the 2 and 5 confusion since the flip is only horizontal (versus vertical).
>
> [1] Bénard, Clément, Brian Staber, and Sébastien Da Veiga. "Kernel Stein Discrepancy thinning: a theoretical perspective of pathologies and a practical fix with regularization." Advances in Neural Information Processing Systems 36 (2024).
>
> [2]Chen, Wilson Ye, et al. "Stein points." International Conference on Machine Learning. PMLR, 2018.
>
> [3] Gorham, Jackson, and Lester Mackey. "Measuring sample quality with kernels." International Conference on Machine Learning. PMLR, 2017.
>
> [4] Riabiz, M., Chen, W. Y., Cockayne, J., Swietach, P., Niederer, S. A., Mackey, L., & Oates, C. J. (2022). Optimal thinning of MCMC output. Journal of the Royal Statistical Society Series B: Statistical Methodology, 84(4), 1059-1081.
>
> [5] Koh, Pang Wei, and Percy Liang. "Understanding black-box predictions via influence functions." International conference on machine learning. PMLR, 2017.
>
> #### Prevent undesired results for the image flip on SVHN
> We didn't do extra effort to validate the image after the horizontal flip. It is due to the observation that model performance can improve significantly with horizontal flip data augmenting during model training, and the model can identify numbers with a horizontal flip. Indeed, humans can identify numbers from a mirror as well. We didn't encounter the 2 and 5 confusion since the flip is only horizontal (versus vertical).

---

> ### Comment · Reviewer_3D8W · 2024-12-02
>
> I thank the authors for their exhaustive responses. I am satisfied with these responses and, while the evaluation setup has its flaws, and that the best setup would be a clinical test, I understand that the current setup is an interesting proxy. I raise my score to 8.

---

### Official Review · Reviewer_mojk · 2024-11-03

**Soundness:** 3
**Presentation:** 3
**Contribution:** 4
**Rating:** 8
**Confidence:** 3

**Summary:**

This paper proposed an example-based explanation method based on kernelized Stein discrepancy (KSD), which can measure the discrepancy between a predictive model and training samples.
In particular, using a model-dependent kernel function used for calculating KSD, the proposed method explains training samples related to the input test sample.
In the experiments on image classification tasks, the proposed method and its variant outperformed the comparing methods in terms of two qualitative metrics.

**Strengths:**

- The proposed method is a good and natural application of KSD, and as a result, it broadens the potential of KSD.
- The derivation of the proposed method is convincing, and thanks to the well-written explanation, it is easy to follow.

**Weaknesses:**

- The proposed method seems a straightforward application of kernel conditional Stein discrepancy (KCSD) [1]. Therefore, the technical novelty of the proposed method is limited.
- Although the experiments were conducted on popular image classification datasets, the evaluation protocol is artificial, especially in evaluating the hit rate.

[1] Jitkrittum, Wittawat, Heishiro Kanagawa, and Bernhard Schölkopf. "Testing goodness of fit of conditional density models with kernels." Conference on Uncertainty in Artificial Intelligence. PMLR, 2020.

**Questions:**

- As related works and comparing methods, Datamodels [2] and its variant, TRAK [3], are not mentioned. Is there a reason why they are not addressed?
- Compared to KCSD, what are the key points of technical novelty in the proposed method?
- In the noise injection setting, it seems that even a simple method that returns the nearest neighbor training sample of the test sample could explain the predictions. Compared to such a baseline, why does the proposed method perform better? Alternatively, can it be demonstrated through experiments that the proposed method is superior?

[2] Ilyas, Andrew, et al. "Datamodels: Predicting Predictions from Training Data." Proceedings of the 39th International Conference on Machine Learning. 2022.

[3] Park, Sung Min, et al. "TRAK: Attributing Model Behavior at Scale." International Conference on Machine Learning. PMLR, 2023.

---

> ### Author Response · Authors · 2024-11-18
>
> Thank you very much for your feedback. Below is our response to the concerns and questions you raised.
>
> #### **Novelty Recap:**
> We want to highlight that applying the Kernelized (Conditional) Stein Discrepancy (KSD) to the prediction explanation itself is novel to the problem. As far as we know, HD-Explain is the only approach that adapts KCSD for this purpose. Adaptation of KCSD for explanation purposes is not a trivial effort. In particular, in a goodness-of-fit context, KSD is to quantify the discrepancy between a fitted distribution and the underlying data distribution. The use of KSD in goodness-of-fit is limited on the level of the overall discrepancy estimation. In contrast, the HD-Explain zooms in the higher granularity by looking at model-conditioned kernel $\kappa_\theta$ among each pair of data, which is further used as an intrinsic score of data influence. We delve deeper into the composition of KSD with greater granularity, focusing on the pairwise relationships between data samples through the Stein Kernel that were never explored in the previous work.
>
> #### **Evaluation Setup:**
> Evaluating the performance of a post-hoc sample-based explanation approach has been a long-lasting challenge in the literature. Previous research [3] often falls back to qualitative evaluation only by showing images as pos/neg influencers to the prediction without further quantitative analysis. Such an evaluation method is subjective and may lead to disagreement among human auditors. To complement such a shortage, some works [4] may use data debugging metrics as a side quantitative support to show the performance of their proposed explanation approach. However, those methods are questionable in terms of whether they quantify the performance of the explanation or data debugging ability (a good data debugging approach is not necessarily a good prediction explanation approach). We aim to avoid falling back on qualitative evaluations and seek a more straightforward alternative to data debugging. Since the sample-based explanation is essentially a ranking problem (where the explanation algorithm should return a ranked list of training samples as high influencers), we adapt classic information retrieval metrics (e.g., hit rate and coverage), which offer a more robust evaluation by eliminating uncontrolled variables that could affect the results.
>
> #### **Rational of Including noise injection in the experiments:**
> While the setting looks trivial with the first glance, it turns out that existing solutions (the well-known baselines) produce only 10\% - 50\% coverage -- many test points receive the same set of explanations, disregarding their unique characteristic. This interesting observation suggests the weakness of the existing approaches on such simple cases where it can happen in practice -- find typical cases with exactly the same symptoms as the new case (in a noise level difference).

---

> ### Author Response · Authors · 2024-11-18
> **Official Comment by Authors (Pt. 2)**
>
> #### **Why data modeling and TRAK are not comparable:**
>
> Datamodeling and TRAK are not designed as a model explanation approach but a data attribution estimation approach, where their objective is to obtain the data importance to a prediction task by minimizing the LDS loss directly; **they do not need a pre-trained machine learning model in their optimization process**. The outcome will be the data attribution list that might contribute to a prediction for *a family of models (e.g. particular neural network architecture)* rather than any particular pre-trained model. Using data modelling objective as example:
> $$\tau_{\text{dm}}(z) \coloneqq \min_{\beta \in \mathbb{R}^n}
>   \frac{1}{m} \sum_{i=1}^m ({{\beta}^\top 1_{S_i} - f(z;{\theta^*(S_i)})}^2 + \lambda \|\beta\|_1$$
> There is no pre-trained model involved at all. Note, $f(z;{\theta^*(S_i)})$ is the model trained on a subset of data $S_i$.
>
> For data modelling [2] specifically, as the authors of the data modelling paper said in their following work (section 2.2 [1]) “**Unfortunately, however, estimating accurate linear predictors requires thousands of samples $(Sj , f(z; θ ⋆ (Sj )))$. Since each sample involves training a model from scratch, this direct estimator can be expensive to compute in large-scale settings**”, the data modelling method is computationally very heavy and it seems not a good idea to use them for the explanation purpose in practice.
>
> Similarly, for TRAK, as the authors of TRAK stated in their paper [1] section 3.2 Steps 1 to 4, the TRAK algorithm aims to advance the data attribution estimation (not explanation). The setting allows them to make multiple aggressive approximations that exceed the model explanation approach, combining 1) linear Taylor expansion, 2) random projection, and 3) Newton approximation. To stabilize the data attribution outcome, in step 4, TRAK asks to train the model multiple times to get an average value. As such, we cannot treat TRAK as a valid baseline to compare.
>
> We have updated our paper draft and included more data attribution descriptions in Appendix Section K.
>
> [1] Park, Sung Min, et al. "TRAK: Attributing Model Behavior at Scale." International Conference on Machine Learning. PMLR, 2023.
>
> [2] Ilyas, Andrew, et al. "Datamodels: Predicting Predictions from Training Data." Proceedings of the 39th International Conference on Machine Learning. 2022.
>
> [3] Yeh, Chih-Kuan, et al. "Representer point selection for explaining deep neural networks." Advances in neural information processing systems 31 (2018).
>
> [4] Koh, Pang Wei, and Percy Liang. "Understanding black-box predictions via influence functions." International conference on machine learning. PMLR, 2017.

---

> > ### Comment · Reviewer_mojk · 2024-11-27
> >
> > I appreciate the detailed explanation provided regarding the novelty and evaluation of your proposed approach. However, I am not yet fully convinced by the reasons for not comparing your method with Datamodels and TRAK.
> >
> > Specifically, you mention that “they do not need a pre-trained machine learning model in their optimization process.” Could you clarify what this means in the context of your argument? Does your proposed method rely on a pre-trained machine learning model for its functionality?
> >
> > Additionally, you note that “TRAK asks to train the model multiple times to get an average value,” but this alone does not seem to justify why TRAK cannot serve as a valid baseline for comparison.

---

> ### Author Response · Authors · 2024-11-28
>
> Thank you for your feedback.  We glad that we have addressed partial of your concerns.
>
> To further address your comments about the comparing HD-Explain with Data modelling approaches, we made further analysis as stated below, where we included our preliminary experimental results (which will be included in the revision) for your reference.
>
> ### **Prediction Explanation vs Data Modelling**
>
> ##### **Objective Difference:**
> We would like to restate the difference between prediction explanation and data modelling.
> Prediction explanation is to explain the prediction of test data point $x_{\text{test}}$ given a **trained** model (e.g. a classifier) $f_{\theta^*}(x)$, where the model parameter ${\theta^*} = {\theta^*}(S)$ is trained on training dataset $S$ and fixed during inference time and explanation time. Specifically, the explanation algorithm takes an input test data point $x_{\text{test}}$ and a **trained** model as input and produces a prediction explanation for $\hat{y}=f_{\theta^*}(x_{\text{test}})$.
>
> Data modelling approaches (e.g. Datamodels and TRAK), on the other hand, do not work on explaining a **trained** model. Instead, it needs either a parameterized function family $f_{\theta}$ (where $\theta$ denotes free parameters that are not trained). Using Datamodels as an example
>
> $$\tau_{\text{dm}}(x_{\text{test}}) \coloneqq \min_{\beta \in \mathbb{R}^n}
>   \frac{1}{m} \sum_{i=1}^m ({{\beta}^\top 1_{S_i} - f(x_{\text{test}};{\theta^*(S_i)})}^2 + \lambda \|\beta\|_1$$
>
> There is NO **trained model** involved at all. Note, $f(x_{\text{test}};{\theta^*(S_i)})$ is a model trained on a **subset of data $S_i$** (not the trained model). It reflects the data modelling approach's intention of looking to reveal data's attribution as an analytical tool.
>
> Similar to TRAK. If the reviewer has the opportunity to run the TRAK algorithm as shared by its authors [2], you will see the instructions as below
> 1. "First, you need **models** to apply `TRAK` to. You can either use the script below to train **three** ResNet-9 models on `CIFAR-10` and save the checkpoints (e.g., `state_dict()`s)"
> 2. "a  `model`  (a  `torch.nn.Module`  instance) — this is the **model architecture/class** that you want to compute attributions for. Note that this model you pass in **does not need to be initialized** (we'll do that separately below)."
>
> Both of the statements tell us that the data attributions' intention is not to explain the prediction for a given **trained** model but an **architecture/class**.
>
> ##### **Empirical Justification:**
> We understand that the reviewer may make this comment due to curiosity about possible "comparability" regardless of their conceptual difference. To this end, we did a quick experiment on CIFAR-10, as shown below.
>
> |   Methods   | Coverage | Hit Rate | Execution Time |
> |:-----------:|:--------:|:--------:|:--------------:|
> | HD-Explain* |   0.23   |   0.01   |      4.73      |
> |  HD-Explain |   0.92   |   0.99   |      13.04     |
> |      IF     |   0.14   |   0.01   |     200.86     |
> |     RPS     |   0.12   |   0.01   |      2.12      |
> |    TRAK*    |   0.48   |   0.06   |     1209.38    |
> |   TracIn*   |   0.14   |   0.01   |      43.71     |
>
>
> Here, we see that TRAK performs better than RPS, IF, and TracIn, consistent with the TRAK paper's experimental results [1]. We note that HD-Explain performs better than TRAK.
>
> In terms of computation, the TRAK will incur **far higher computation costs** than the well-known prediction explanation approaches we compared in this paper. It reflects our previous justification of the difference in motivation between prediction explanation and data attribution.
>
> ##### **Application Perspective:**
> HD-Explain, as a data-centric prediction explanation tool, is to help verify a **trained** model's prediction given a list of explanations during inference time. Its application is more often to the **deployed** AI systems in practice where practitioners do not modify the trained model.
>
> Data modelling approaches to help analyze the potential data attribution to **a model architecture/class** as a debugging tool. It is more suitable for use during model training time before deployment.
>
> Given the reviewers' comments, we will include the above empirical comparison in our paper revision later, along with the above discussion.
>
>
>
> [1] Park, Sung Min, et al. "TRAK: Attributing Model Behavior at Scale." International Conference on Machine Learning. PMLR, 2023.
>
> [2] https://trak.readthedocs.io/en/latest/quickstart.html

---

> > ### Comment · Reviewer_mojk · 2024-11-29
> >
> > Thank you for your response. I understood the difference between the proposed method and TRAK clearly. Also, the additional experimental results clarified the advantages of the proposed method compared to TRAK. My concerns were alleviated, therefore, I will raise the score.

---

### Official Review · Reviewer_54VY · 2024-11-04

**Soundness:** 3
**Presentation:** 2
**Contribution:** 2
**Rating:** 6
**Confidence:** 3

**Summary:**

The paper presents HD-Explain, a data-centric, example-based explanation method leveraging Kernelized Stein Discrepancy (KSD). The authors argue that existing example-based methods for explaining predictions, such as Influence Functions and Representer Point Selection, suffer from high computational costs and coarse-grained explanations. HD-Explain addresses these challenges by using KSD to establish a fine-grained, efficient link between training samples and model predictions. Through empirical evaluations on datasets like CIFAR-10 and medical imaging, the paper shows that HD-Explain outperforms traditional methods in terms of explanation precision, consistency, and computational efficiency.

**Strengths:**

1. Applying Kernelized Stein Discrepancy to interpretability is novel and conceptually interesting. Unlike other methods that can struggle with memory or computation limits in large models, HD-Explain maintains a low computational cost, which support its use in high-stakes applications where explanation and efficiency are essential. The focus on healthcare datasets underscores the authors’ intent to target real-world, interpretable machine learning needs.

2.   The paper attempts to quantify explanation performance with metrics like Hit Rate, Coverage, and Run Time. These metrics provide a foundation for benchmarking HD-Explain against other methods in terms of granularity, diversity, and efficiency.

**Weaknesses:**

1. The proposed method provides a new use case of KSD, which has been used in several other cases, such as goodness-of-fit applications, which are quite related to the new use case. The paper extends KSD to explainability, but the novelty is limited to an application shift, applying an existing statistical method to a new context rather than developing a new technique for explanation in machine learning.

2. The paper explains that KSD is close to zero if a model fits the data well, which is shown in previous studies. However, I feel there is a gap of why KSD is a good choice for model explanation, specifically, selecting training samples given a test sample. Also there are other discrepancy measures, such as Maximum Mean Discrepancy (MMD), it's unclear to see why KSD is a better choice for this task among other discrepancy measures. Without testing these alternatives, the choice of KSD appears arbitrary, which weakens the novelty claim.

3. In addition to the newly proposed evaluation metrics, existing metrics such as Linear Datamodeling Score (LDS) seem to be widely used in training data attribution methods. LDS was not reported in this proposed paper.

**Questions:**

Suggestions:

1. The authors might consider adding an algorithm to show how the proposed method works.

2. The authors are suggested to report LDS, if possible.

---

> ### Author Response · Authors · 2024-11-18
>
> Thank you very much for your feedback. Below is our response to the concerns and questions you raised.
> #### **Novelty:**
> We would like to highlight the novelty of this paper as follows:
> 1. HD-Explain is the very first training-sample-based explanation method that 1) does not leverage model parameter perturbation so far in the literature and 2) shows intuitively understandable explanation samples (as shown in Figures 3, 4, 8).
> 2. HD-Explain is the very first explanation method leveraging classic statistical discrepancy with kernel tricks. While the extension of Stein discrepancy can be traced back to 1970 and has recently been used for goodness of fit, the statistical tool has never been explored to explain machine learning models, as we showed in paper Section 2.1. Adapting KSD for explanation purposes is a non-retrieval task where it zooms in on the pairwise training data that is never examined in the goodness of fit literature.
> 3. Standard KSD in the goodness of fit literature is commonly limited to the generative model, which focuses on examining the fitness of $P(\mathbf{x})$. In HD-Explain, we made substantial adjustments to adapt it to consider a whole data point $P(\mathbf{x}, y)$ by re-defining the corresponding kernels.
> 4. HD-Explain is a prediction explanation method; the research field is very different from the goodness of fit literature.
> As the reviewer stated in the next question about the rationale of having KSD in the explanation, the idea (Explanation with KSD) behind HD-Explain is not trivial.
>
> #### **How KSD support model explanation:**
>
> The great advantage of KSD is the direct calculation of Discrepancy between discrete (data q) and continuous (model p) distributions by the finite sum of kernels. It quantifies the disparity between the two distributions as an expectation of Stein kernels
> $$ \mathbb{S}(q,p) = \mathbb{E}_{x,x'\sim q} [\kappa_p(x,x')], $$
> where the Stein operator augmented kernel $\kappa_p$ uniquely defines pairwise data correlation within the context of a trained model, which can be viewed as a data mesh of the trained model. In other words, the kernel $\kappa_p$ is the model-augmented data correlation that can reflect the tie between a pair of data points. This observation inspired the proposed HD-Explain. By leveraging this property, we can achieve two objectives: 1) pinpoint a subset of training data points that offer the best predictive support for a given test point and 2) detect potential distribution mismatches among the training data points.
> We believe this is a valuable point to be included in the paper. Hence, we have updated our paper draft by including more discussion about this topic. Please see section 4.4 on the revised paper draft.
>
> #### **Possibility of using other discrepancy definition:**
>
> KSD has unique property that allow calculation of discrepancy between a discrete and continuous distribution where the normalization factor for the continuous distribution is not feasible to calculate. In terms of not using MMD, we note that the samples from the continuous distribution are not available which is the requirements for the MMD. In fact, KSD can reduce to a MMD under certain conditions as described [1]. Other well-known diversity definitions, such as KL divergence, are not applicable in measuring the discrepancy between data and modeled distributions since they do need prior knowledge of the data or modeled distribution in a well defined form, which is not accessible here. E.g. Gaussian.
>
> #### **Include Algorithm:**
> Thanks for this great suggestion, we have updated our paper draft and included a algorithm section in appendix (Section L) for your review.

---

> ### Author Response · Authors · 2024-11-18
> **Official Comment by Authors (Pt. 2)**
>
> #### **Why not using LDS as a metric:**
> As mentioned by the reviewer, the LDS is a metric for data attribution estimation, not for model prediction explanation specifically. We carefully analyzed the feasibility of including this metric but realized the substantial difference between the two research fields. We have updated our paper draft and included a new section (Section K) to discuss the difference.  However, to respectfully address the reviewer’s comment regarding this metric, we followed the LDS evaluation approach proposed in the paper and conducted an experiment on the CIFAR-10 dataset. The average and standard deviation of the LDS are presented in the table below.
> We have conducted several rounds of experiments to ensure that LDS is not a reliable evaluation metric and demonstrate its variation due to the randomness in sampling the subsets of the data. Here is the provided result for the experiment on the CIFAR10 dataset .
>
> | Method | LDS |
> |-------------|----------------|
> | **HD-EXPLAIN** | 0.0771 ± 0.04 |
> | HD-EXPLAIN* | 0.00093 ± 0.05 |
> | IF | 0.00408 ± 0.03 |
> | RPS | 0.02309 ± 0.03 |
> | TracIn | 0.01140 ± 0.03 |
>
> While the metric shows our HD-Explain does better than the baselines, we are not confident with this results given too much variables in the LDS metric, including retraining multiple models on subset of the training dataset. It introduced much noise and making this metric highly unreliable for evaluating prediction explanation algorithms. In particular, from the provided results, we can observe that LDS is notoriously unstable. It requires heavy stress test (large number of subset samples, large number of test sets and ) to ensure its soundness (as proposed in the original paper [1] estimating accurate linear predictors requires thousands of samples, and they conducted the experiment on 2000 test sets for 100 subsets of data)
>
> [1] Park, Sung Min, et al. "TRAK: Attributing Model Behavior at Scale." International Conference on Machine Learning. PMLR, 2023.

---

> ### Author Response · Authors · 2024-11-25
>
> Dear Reviewer,
>
> Thank you for your feedback on our manuscript. We believe we have addressed all your comments. With the rebuttal deadline approaching, we would greatly appreciate a discussion regarding our responses. Please let us know if there are any points that require further clarification.
>
> Authors

---

### Meta-Review · Area_Chair_jYyD · 2024-12-17

**Metareview:**

The paper presents a framework for data-centric example-based model explanation using Kernelized Stein discrepancy. The main claimed contribution is that the proposed method outperforms previous work in terms of explanation precision, computational efficiency and consistency. Overall, the reviewers believe the paper presents a conceptually novel idea with good experimental support. Several issues were raised during the discussion period such as the technical novelty of the paper, comparisons to other approaches (such as TRAK)  and the quality of the presentation. I believe the authors have addressed these issues satisfactorily. Therefore, I recommend acceptance.

**Additional Comments On Reviewer Discussion:**

Several issues were raised during the discussion period such as the technical novelty of the paper, comparisons to other approaches (such as TRAK)  and the quality of the presentation. I believe the authors have addressed these issues satisfactorily.

---

### Decision · Program_Chairs · 2025-01-22

Accept (Poster)